

# A climate-driven, altitudinal transition in rock glacier dynamics detected through integration of geomorphological mapping and InSAR-based kinematics

Aldo Bertone [1], Nina Jones [2], Volkmar Mair [3], Riccardo Scotti [4], Tazio Strozzi [2], Francesco Brardinoni [1]

[1] Department of Biological, Geological and Environmental Sciences, University of Bologna, Bologna, 40126, Italy
[2] Gamma Remote Sensing, Gümligen, 3073, Switzerland
[3] Ufficio Geologia e Prove Materiali, Provincia Autonoma di Bolzano, Cardano, 39053, Italy
[4] Servizio Glaciologico Lombardo, La Valletta Brianza, 23888, Italy

*Correspondence to*: Francesco Brardinoni (francesco.brardinoni@unibo.it)

**Abstract.** In dry southwestern South Tyrol, Italy, rock glaciers are dominant landforms of the high-mountain cryosphere. Their spatial distribution and degree of activity hold critical information on the past and current state of discontinuous permafrost, and consequently on response potential to climate warming. Traditional geomorphologic mapping, however, owing to the qualitative expert-based nature, typically displays a high degree of uncertainty and variability among operators with respect to the dynamic classification of intact (permafrost bearing) and relict (permafrost devoid) rock glaciers. This limits the reliability of geomorphologic rock glacier inventories for basic and applied purposes. To address this limitation: (i) we conduct a systematic evaluation of the improvements that InSAR-based information can afford to the detection and dynamic classification of rock glaciers; and (ii) build an integrated inventory that wishes to combine the strengths of geomorphologic- and InSAR-based approaches. To exploit fully InSAR-based information towards a better understanding of the topo-climatic conditions that sustain creeping permafrost, we further explore how velocity and the spatial distribution of moving areas (MAs) within rock glaciers may vary as a function of simple topographic variables known to exert first-order controls on incoming solar radiation, such as elevation and aspect. Starting from the compilation of a geomorphologic inventory (n = 789), we characterize the kinematics of InSAR-based MAs and the relevant hosting rock glaciers on thirty-six Sentinel-1 interferograms computed over 6- through 342-day baselines in the 2018-19 period. With respect to the original inventory, InSAR analysis allowed identifying 14 previously undetected rock glaciers. Further, it confirmed that 246 (76%) landforms, originally interpreted as intact, do exhibit detectable movement (i.e., ≥1 cm yr$^{-1}$), and that 270 (60%) of the relict labelled counterparts do not, whereas 144 (18 %) resulted kinematically undefined due to decorrelation. Most importantly, InSAR proved critical for reclassifying 121 (15%) rock glaciers, clarifying that 41 (13%) of those interpreted as intact, do not exhibit detectable movement, and that 80 (17%) of the original relict ones do actually move. Reclassification, by increasing the altitudinal overlap between intact and relict rock glaciers depicts a broad transition belt in the aspect-elevation space, the amplitude of which varies from as little as 50 m on west facing slopes to a maximum of 500 m on easterly ones. This finding deteriorates the significance of elevation and aspect as topographic proxies for modelling permafrost occurrence, and highlights the importance of using InSAR for informing such models. From a process-oriented standpoint,



InSAR information proves fundamental for imaging how this altitudinal transition manifests through changing rates and

styles of rock glacier surface deformation. Specifically, we find that as rock glaciers move faster, an increasingly larger proportion of their surface becomes kinematically involved (i.e., percent MA cover), and that this proportion increases with elevation up to the 2600-2800 m, beyond which an inflection occurs and consistent average values are attained. Considering that the inflection falls between the -1°C and -2 °C MAAT – the lower boundary for discontinuous permafrost – and is independent of slope gradient, we conclude that this altitudinal pattern represents a geomorphic signature: the dynamic

expression of increasing permafrost distribution (i.e., from sporadic to discontinuous), until optimal thermal conditions are reached.

## 1 Introduction

In the last four decades, atmospheric temperature rise has led to rapid glacial retreat and permafrost degradation in high mountain environments, which has promoted slope instability (e.g., Huggel et al., 2015; Kos et al., 2016; Frattini et al., 2016;

Schlögel et al., 2020) and reduction in water storage potential (e.g., Azocar and Brenning, 2010; Jones et al., 2018). To assess relevant geohazards and adaptation measures in relation to climate change scenarios, a comprehensive characterization and monitoring of the alpine cryosphere is needed. While glacier change is being successfully monitored worldwide through analysis of satellite imagery (e.g., Paul et al., 2004; Bolch et al., 2010), evaluating the spatial distribution of mountain permafrost and its ongoing degradation across regions is more challenging. This task may be tackled at the

regional scale only indirectly, by considering the spatial distribution of rock glaciers – complex depositional landforms that mainly develop through the former or current creep of perennially frozen ice-rich debris (permafrost), where creep refers to the variable combination of both internal deformation and shearing at depth – which are regarded as unambiguous morphological evidences of permafrost-bearing conditions (e.g., Haeberli et al., 2006; Harris et al., 2009). Operational steps entail the compilation of a rock glacier inventory, the dynamic classification of each rock glacier i.e., active, inactive (also

termed transitional), or relict (Barsch, 1996), and the relevant modelling based on the spatial distribution of intact (i.e., active and inactive) rock glaciers and available empirical measurements (e.g., Imhof, 1996; Lambiel and Reynard, 2001; Boeckli et al., 2012; Schmid et al., 2015).

Traditionally, inventories are compiled via identification, manual delineation and dynamic classification of rock glaciers on optical imagery, complemented by some confirmatory fieldwork (e.g., Scotti et al., 2013; Falaschi et al., 2015; Onaca et al.,

2017; Wagner et al., 2020; Johnson et al., 2021). Owing to the variability and complexity of rock glacier typology and morphology, compiling a "geomorphologic" rock glacier inventory requires some expert-based interpretation of landforms. Accordingly, the completeness and reliability of an inventory depends not only on image quality, but also on the operator's mapping approach and experience. In this context, a recent comparative study has indicated varying degree of subjectivity in inventories compiled by different operators for a given study area, highlighting high variability associated with the

morphological discrimination between intact and relict landforms (Brardinoni et al., 2019). This represents a practical



limitation for evaluating permafrost distribution and geohazard potential, as the altitudinal transition between intact and relict rock glaciers typically sits at the fringe of permafrost-bearing terrain.

While this limitation can be overcome locally through ground confirmation and variable combinations of topographic, photogrammetric, geodetic and geophysical surveys (e.g., Konrad et al., 1999; Delaloye et al., 2008; Bodin et al., 2009; Vivero and Lambiel, 2019; Fey and Krainer, 2020), when wishing to upscale the assessment of rock glacier activity over entire basins, or regions, satellite Synthetic Aperture Radar Interferometry (InSAR) may prove fundamental. As a consolidated technique for detecting and mapping land surface deformation at suitable spatial and temporal resolution (Rosen et al., 2000), InSAR warrants an objective kinematic characterization (hence dynamic classification) of rock glaciers over large areas (e.g., Liu et al., 2013; Necsoiu et al., 2016; Wang et al., 2017; Bertone et al., 2019; Brencher et al., 2021; Reinosch et al., 2021). In this context, we argue that the kinematic approach – developed by Barboux et al. (2014) and refined by Bertone et al. (2022) – entailing the detection and delineation of moving areas (i.e., areas of detectable surface deformation on wrapped interferograms), besides elucidating which rock glaciers move, and consequently bear permafrost, may prove strategic for documenting where about the rock glacier (e.g., the main front, the rooting zone, or the entire landform) and in what proportion surface deformation occurs. Following this logic, this type of spatially-distributed information may open opportunities to improve our understanding of the topo-climatic conditions that control rock glacier activity (and ultimately permafrost persistence) in a mountain landscape. Although the application of InSAR technology to rock glacier inventories holds straight forward advantages, amply demonstrated for single, a cluster, or many rock glaciers (e.g., Barboux et al., 2014; Strozzi et al., 2020; Lambiel et al., 2023; Bertone et al., 2023), a systematic and quantitative evaluation of the improvements afforded to a traditional "geomorphologic" inventory, encompassing both intact and relict landforms over broad spatial scales, is missing.

To address this gap, following operational guidelines on the InSAR-based kinematic characterization of rock glaciers jointly proposed by ESA Permafrost CCI and IPA Action Group on rock glacier inventories (Bertone et al., 2022), we wish to integrate geomorphologic and InSAR-based inventorying approaches in selected valleys of western South Tyrol, where rock glacier occurrence is overwhelming. Therein, starting from the compilation of a geomorphologic inventory, we aim to: (i) characterize the kinematics of InSAR-based moving areas and relevant hosting rock glaciers; and (ii) evaluate InSAR-derived improvements, in terms of inventory completeness and uncertainty reduction, in the detection and dynamic classification of rock glaciers. To fully exploit InSAR-based information towards a better understanding of the current topo-climatic conditions associated with creeping permafrost, we further aim to explore in which way the velocity and spatial distribution of moving areas within rock glaciers may vary as a function of simple topographic variables known to exert first-order controls on incoming solar radiation and ground temperature, such as elevation and aspect. To pursue these objectives, we use Sentinel-1 interferograms over the 2018-19 period, as they warrant consistent and freely available acquisitions with short (i.e., 6 days) repeat-time intervals suitable to examine systematically rock glacier surface deformation over large areas.





## 2 Study area

This work is concerned with rugged mountain terrain of the north-eastern portion (970 km$^2$) of the Ortles-Cevedale massif in the Autonomous Province of Bozen, Central-Eastern Italian Alps (46°31' N, 10°50' E, **Fig 1a**). It comprises the southern side of lower Vinschgau/Venosta Valley as well as five tributary valleys, including Ultental/Ultimo Valley, Martelltal/Martello Valley, Laasertal/Lasa Valley, and Suldental/Solda Valley. Elevation ranges from about 500 m a.s.l. at Ultimo Valley outlet, up to 3905 m of Mount Ortles. Bedrock geology is dominated by metamorphic lithologies (chiefly

paragneiss, micaschists, and orthogneiss), with granite outcropping locally in lower Martello Valley, and limestones and dolostones in upper Solda Valley (Keim et al., 2013).

Climate is dry, with mean annual precipitation ranging from 506 mm (1921-2020) at Schlanders/Silandro (698 m a.s.l.) on the Venosta valley floor, to 779 mm (1972-2020) at Zufritt/Gioveretto Dam (1851 m a.s.l.) in upland valleys. According to Permanet modelling (www.permanet.eu) and field-based evidences, discontinuous mountain permafrost roughly occurs

above threshold elevations varying between 2300 m and 2700 m a.s.l., depending on topographic aspect and microclimatic, site-specific conditions (Boeckli et al., 2012). This elevation belt roughly agrees with the lower boundary for discontinuous permafrost occurrence, as constrained by the -1°C and -2 °C MAAT (mean annual air temperature) envelope (i.e., Haeberli, 1983; Haeberli et al., 1989), which in southwestern South Tyrol, based on regional climatic characterization (1981-2010), sets between 2595 m and 2760 m a.s.l. (www.alpenklima.eu).

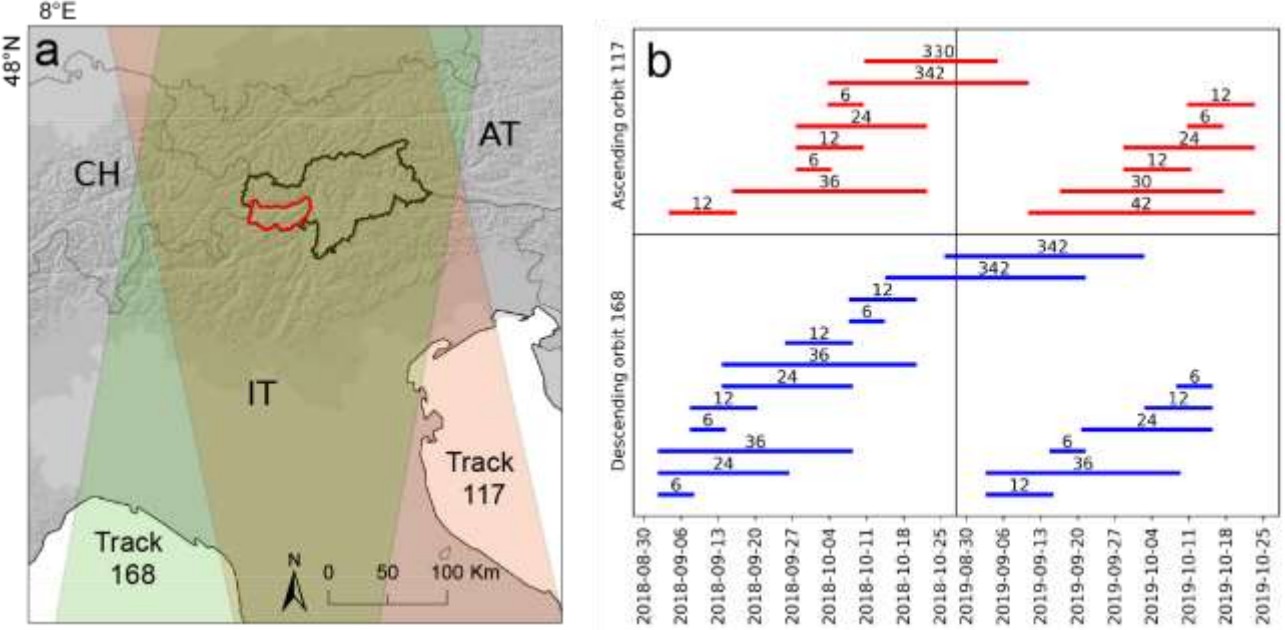


Figure 1. (a) Autonomous Province of Bozen (black linework) and Southern Venosta Valley (red linework), including footprints of Sentinel-1 ascending orbit #117 and descending orbit #168 used for InSAR analysis in this study. (b) Summary of the interferograms used for the mapping and kinematic characterization of moving areas. Horizontal bars represent the time intervals of the wrapped interferograms expressed in number of days. Hillshade from European Environment Agency

(2004). https://data.opendatascience.eu/geonetwork/srv/api/records/84036394-19fc-466f-bc4b-b0748d5d29f4.



## 3 Data collection and analysis

### 3.1 Rock glacier inventory and degree of activity

The geomorphologic rock glacier inventory in south-western South Tyrol is part of a broader mapping effort conducted
across the entire Autonomous Province of Bozen. The identification, mapping and dynamic classification of each rock
glacier relies primarily on the visual interpretation of 0.2-m gridded, optical imagery (i.e., 2014, 2017, and 2020) and a 2.5-
m LiDAR-derived hillshade raster (i.e., 2006), and where available, on information drawn from local reports on road closure
or damage to infrastructure associated with the advance of rock glacier fronts. Landform delineation typically starts at the
base of the rock glacier's front and proceeds through the lateral margins, up to the lower end of the rooting zone depression,
which in this study is excluded from the rock glacier polygon (e.g., Brardinoni et al., 2019). Dynamic classification, which
accounts both for the presence/absence of ice-rich debris (i.e., permafrost) and rock glacier's degree of mobility, follows the
classical three-part scheme summarized by Barsch (1996) and as such distinguishes among active, inactive, and relict
landforms. Accordingly, active rock glaciers exhibit downslope movement (i.e., $\geq 1$ cm yr$^{-1}$) at their fronts and over large
parts of their surface. Downslope movement is considered to be chiefly the result of ice content sufficient to sustain creep
and internal shearing. Inactive (or transitional, according to RGIK (2022)) rock glaciers, which in principle would still
contain permafrost, even though in a lesser amount, do not move at their front (i.e., $< 1$ cm yr$^{-1}$) and may exhibit subsidence,
but could still support downslope shearing and displacement on their upper portions (i.e., Barsch, 1996, p. 190). Relict rock
glaciers, as a result of exhausted permafrost content do not bear any kind of surface deformation.

Considering the high variability (i.e., the little consensus among different operators) associated with the morphologically-
based dynamic classification of inactive rock glaciers (i.e., Brardinoni et al., 2019; cf., their Fig 11), in this work, we will
merge active and inactive landforms in the so-called intact category (Haeberli, 1985; Barsch, 1996) and will evaluate their
spatial distribution against that of relict counterparts. Following this logic, although permafrost occurrence can be
unambiguously determined only through direct coring or (indirectly) via geophysical prospecting, in this contribution we
will assume that intact rock glaciers bear permafrost and that relict rock glaciers do not. Similarly, in the next section, we
will regard InSAR-based surface displacement $\geq 1$ cm yr$^{-1}$ detected within rock glaciers' morphological footprints (i.e.,
"moving" rock glaciers) as diagnostic of permafrost occurrence, and equivalent to an *intact* dynamic state. Vice versa, the
lack of movement $\geq 1$ cm yr$^{-1}$ (i.e., "not moving" rock glaciers) – excluding cases affected by decorrelation – will be
regarded as conditions compatible with a *relict* dynamic state. As will become clear, the 1 cm threshold is dictated by the
maximum time interval considered in this study, which is one year.

### 3.2 InSAR-based kinematic characterization of moving areas and rock glaciers

The kinematic characterization of rock glaciers follows the specifics proposed by the International Permafrost Association
(IPA) Action Group on rock glacier inventories and kinematics (2018–2023), with the support of the European Space

Agency (ESA) Permafrost Climate Change Initiative (CCI) (Bertone et al., 2022). Accordingly, the methodology entails the identification, manual delineation and kinematic classification of *moving areas* on Sentinel-1 wrapped interferograms (**Fig**

**2**), where a moving area is a portion of detectable surface deformation within a hosting rock glacier polygon, as outlined in the geomorphologic inventory (**Fig. 3**). Subsequently, moving areas are used to classify the relevant hosting rock glaciers according to specific kinematic classes. It follows a detailed description of the methodological steps involved (**Fig. 2**).

**Figure 2.** Schematic flow diagram illustrating the data sources and the methodological steps involved in the dynamic
classification of rock glaciers obtained through integration of the relevant geomorphologic information and InSAR-based kinematics. See text for detailed description of methodology.

### 3.2.1 Interferometric data processing

In this work, InSAR analysis of Sentinel-1 acquisitions is performed using the Gamma Software along tracks 117 (ascending) and 168 (descending) (**Fig. 1**). Since snow cover represents a severe limitation to satellite SAR data analysis
(Klees and Massonnet, 1998), acquisitions of interest are restricted to snow-free periods only i.e., from the beginning of



September to the end of October in 2018 and 2019 (**Fig. 1b**). In the Alps, this seasonal timing roughly corresponds with the period of maximum rock glacier displacement within the year (Delaloye and Staub, 2016; Wirz et al., 2016), and indirectly ensures highest detectability rates across the population of favourably exposed rock glaciers. Collectively, we have processed 28 images in Interferometric Wide (IW) mode with a 250 km swath at 5 m by 20 m spatial resolution, Single Look

Complex (SLC) product type and Vertical transmit Vertical receive (VV) polarization.

Differential interferograms are computed with 1 look in range and 4 looks in azimuth, by combining pairs of images with time intervals of 6, 12, 24, 30, 36, 42, 330 and 342 days. Areas affected by foreshortening, layover and shadow are masked out (Klees and Massonnet, 1998). Topographic phase corrections on interferograms, as well as data geolocation are conducted using a 2.5 m gridded LIDAR DTM (Bamler and Hartl, 1998; Yague-Martinez et al., 2016). Ultimately, we have

computed 32 wrapped interferograms (**Fig. 1b**).

Wrapped interferograms provide surface displacement information. However, some limitations apply (Klees and Massonnet, 1998; Yague-Martinez et al., 2016). First, displacements between adjacent pixels higher than half of the wavelength (i.e., 2.77 cm for Sentinel-1), are not measurable because the interferometric phase signal becomes ambiguous, generating decorrelation (Massonnet and Souyris, 2008). To minimize this issue, interferograms with a range of temporal baselines are

used. Second, artefacts due to uncompensated atmospheric delays (Yu et al., 2018) and decorrelation, or phase bias, due to changes in physical properties of the surface (e.g., vegetation, snow, and soil moisture) (Klees and Massonnet, 1998; Zwieback et al., 2016) can prevent the detection of movement. Third, the orientation of a rock glacier's main flow line, with respect to the satellite's flying direction, represents a major limitation to detecting ground deformation. Specifically, since InSAR is sensitive only to the component of the three-dimensional surface deformation projected along the radar look

direction (i.e., the so-called Line Of Sight, LOS), the greater the geometric difference between the true three-dimensional flow direction of a rock glacier and the LOS, the greater the underestimation of the one-dimensional LOS measurements (Barboux et al., 2014; Klees and Massonnet, 1998). Assuming the downslope direction being representative of the real three-dimensional movement of a rock glacier (Barboux et al., 2014; Liu et al., 2013), the underestimation may be evaluated by computing the $\alpha$ angle between the downslope direction and the LOS (equation 1):

$$U = (1 - \cos\alpha) \cdot 100 \qquad (1)$$

Following this logic, underestimation of ground deformation increases progressively as one examines rock glaciers approaching northerly or southerly slope aspects, due to downslope movements directed about parallel to the satellite's flight direction (i.e., perpendicular to LOS direction), which InSAR cannot detect. In this context, concurrent consideration of both ascending and descending geometries warrants greater flexibility of acquisition and minimizes the number of unfavourably

exposed landforms. Nevertheless, in rugged alpine terrain rock glacier motion may still be subject to substantial underestimation, or major distortions known as layover or shadowing, due to peculiar topographic configurations (i.e., combinations of slope aspect, gradient, and curvature). We consider InSAR measurements as unreliable when underestimation exceeds 50%, which corresponds to an $\alpha$ angle of 60° (Klees and Massonnet, 1998; Barboux et al., 2014).



**Figure 3.** Sample kinematic characterization applied to two rock glaciers in Ultimo Valley showing (a) initial manual delineation of rock glacier polygons (black linework) on optical imagery; Sentinel-1 interferograms calculated over (b) 6 days (2018/09/28 – 2018/10/04), (c) 12 days (2019/10/11 – 2019/10/23), (d) 24 days (2018/09/28 – 2018/10/22), and (e) 330 days (2018/10/10 – 2019/09/05). In panel f is reported the color-coded scheme used for evaluating phase difference on interferograms. RG2 displays consistent plain pattern across all interferograms, indicating lack of movement above instrumental limit of detection (i.e., < cm yr⁻¹). RG1 displays partial fringe (6 days), complete fringe (i.e., a complete phase cycle) (12 days) and finally decorrelation pattern (24 days and longer) with increasing temporal baseline. The velocity class



of each moving area is estimated by counting the number of phase cycles, converting them into displacement (i.e., one phase cycle in Sentinel-1 corresponds to a displacement of 2.77 cm), and then converting displacement into velocity (red and orange linework in panels e through g). Each rock glacier is finally assigned to a velocity class (green and red centroids)
according to the velocity of the relevant moving area(s), following the classification scheme illustrated in **Fig. 2**. Orthoimage from the Autonomous Province of Bolzano (https://geoportale.retecivica.bz.it/geodati.asp; last access: June 2023).

To disentangle downslope from vertical components of deformation, some studies have projected the measured LOS displacement along the maximum slope direction of a rock glacier, as calculated from a Digital Elevation Model, while others have combined ascending and descending geometries to derive east-west and up-down components, assuming that no
movement occurs along the north-south axis (e.g., Liu et al., 2013; Brencher et al., 2021; Reinosch et al., 2021). In our opinion, both approaches hold intrinsic bias, the magnitude of which is difficult to assess. The former is highly dependent on the quality of the DTM, which in remote mountain areas are typically coarse in resolution (e.g., 30 m). Moreover, downslope creep of coarse debris does not necessarily follow the steepest descent. The latter, by conveying movement along the north-south axis mainly into the up-down component, may lead to unrealistic rates of deformation. For these reasons, in this work
we prefer to stick to displacements along the LOS, and express the relevant reliability based on the orientation of the rock glacier with respect to the LOS.

### 3.2.2 Inventorying moving areas

A moving area is defined as the portion of a rock glacier surface in which the direction and the velocity field of ground deformation are spatially consistent and homogenous. Accordingly, a moving area depicts the rate of movement of a portion
of a given rock glacier, along the one-dimensional LOS. In this study, each moving area is related to a specific (seasonal or annual) time window of observation comprised between 02 Sept 2018 and 23 Oct 2019.

Moving areas are manually mapped on wrapped interferograms (Barboux et al., 2014). Specifically, they are identified by looking at textural image features such as: (i) "fringe patterns" (**Fig. 3b**), expression of detectable displacement; (ii) "plain patterns" (**Fig. 3b** through **3e**), in the absence of detectable displacement; and (iii) "noisy (decorrelated) patterns" (**Fig. 3d**
and **3e**) associated to local ground noise (e.g., vegetation cover) or high (over threshold) rates of displacement (**Fig. 3f**). The rationale for inventorying moving areas is two-fold, they serve for characterizing the degree of activity (i.e., intact or relict) of rock glacier polygons mapped in the geomorphologic inventory, as well as for identifying additional rock glaciers that may have gone undetected during aerial photo interpretation. Following this logic, the visual inspection of interferograms first focuses within the "geomorphologic" rock glacier outlines, and then expands outside. When an "outsider" moving area
is detected, an iterative process starts, and the manual delineation of a new rock glacier polygon is conducted on historical optical imagery and LiDAR-derived hillshades.

To ensure that different operators outline moving areas consistently, as well as to distinguish moving areas from surrounding noise, a minimum area threshold is applied. That is, moving areas need to involve at least 20 adjacent pixels in a gridded interferogram. The outlines are drawn according to the detected fringe pattern, without necessarily fit the entire
geomorphologic outline of the rock glacier. In the present work, this mapping procedure is based on the combined visual





inspection of the 32 interferograms (**Fig. 3b** through **3e**). Outlines are provisionally drawn starting from interferograms associated with shortest time intervals. As longer intervals are considered, moving areas are progressively refined and enriched with additional ones characterized by lower rates of displacement. Incidentally, this redundant multi-temporal approach allows minimizing decorrelation and local artefacts associated with unfavourable atmospheric and/or ground
conditions (Barboux et al., 2014; Yu et al., 2018).

Following outline delineation, a specific velocity class is assigned to each moving area, where a velocity class is meant to reflect the mean movement rate over the 2018-19 study period i.e., not a specific intra-annual variation or an extreme value. Assigned annual velocity classes include: 1-3 cm yr$^{-1}$; 3-10 cm yr$^{-1}$; 10-30 cm yr$^{-1}$; 30-100 cm yr$^{-1}$; and > 100 cm yr$^{-1}$ (**Fig. 2**). Class membership is assigned depending on the change in colour observed on the wrapped interferograms (**Fig. 3b**
through **3e**). A change in colour expressed by fringe(s) corresponds to variations of the interferometric phase, correlated to the ground deformation projected on the LOS direction (Klees and Massonnet, 1998; Strozzi et al., 2020). For example, a complete phase cycle corresponds to a displacement of 2.77 cm (half of one Sentinel-1 wavelength). Velocity is finally computed considering the temporal baseline of the interferogram under examination. The maximum temporal baseline considered in this work is one year. This choice sets the lower limit of velocity detection on Sentinel-1 interferograms to
about 1 cm yr$^{-1}$ (Barboux et al., 2014; Yague-Martinez et al., 2016). That is, in this work we will not able to distinguish between areas with movement < 1 cm yr$^{-1}$ and areas with no motion.

### 3.2.3 Rock glacier kinematic classification and analysis

Moving area characteristics - including extent, velocity class, and temporal baseline - are then used to assign the dynamic state (or degree of activity) to rock glaciers of the geomorphologic inventory (**Fig. 2** and **3e**). In particular, we distinguish
among: (i) *moving* rock glaciers, enclosing one or more moving areas; (ii) *not moving* rock glaciers, which do not contain any detectable moving area (above ≈ 1 cm yr$^{-1}$) on annual interferograms; and (iii) *undefined* rock glaciers, for which reliable kinematic information is not available, due to extended layover, shadowing, atmospheric artefacts, phase bias, or decorrelation. Based on the kinematics of the relevant moving area(s), each moving rock glacier is further assigned to a first-order mean annual velocity category i.e., cm yr$^{-1}$; cm to dm yr$^{-1}$; dm yr$^{-1}$; and dm to m yr$^{-1}$ (**Fig. 2**), which in this paper we
call "kinematic classification". Categorical assignment takes into account the seasonal variability of rock glacier downslope deformation, generally highest during late summer and autumn months (Delaloye and Staub, 2016; Kellerer-Pirklbauer et al., 2018). To a rock glacier that hosts multiple moving areas with diverse kinematics, we assign the median class, while giving more weight to the largest moving area that is closest to the rock glacier front. To evaluate the extent to which rock glacier kinematics may depend on local, first-order topo-climatic conditions, we first examine how the size and velocity of moving
areas (i.e., the rock glacier's kinematic building blocks) vary across elevations and aspects. Subsequently, at a higher hierarchical level, we examine the internal kinematic configuration of each rock glacier, that is, how *total moving area* (i.e., the proportion of a rock glacier footprint that actually moves) varies as a function of elevation, aspect and displacement rate.



## 4 Results

We begin by presenting the dynamic and topographic characterization of the geomorphologic rock glacier inventory. We
continue with the InSAR-based kinematic characterization of moving areas and hosting rock glaciers, then we compare the
geomorphologic and InSAR-based dynamic classification approaches and combine them in a so-called "integrated"
inventory. We finally examine the velocity and spatial distribution of moving areas in relation to elevation, aspect, and size
of the hosting (geomorphologic) rock glacier polygon.

### 4.1 Rock glacier geomorphologic inventory

We have identified and mapped on optical imagery 789 rock glaciers, which occupy a total area of 35.55 km$^2$. Among these,
322 and 457 are interpreted respectively as intact and relict landforms (**Table 1** and **Fig. 4** and **5a**). The former are about one
third smaller than the latter (cf. median size in **Table 1**), and tend to occur at higher elevations. Specifically, median
elevation of intact rock glacier fronts is located about 400 m higher (i.e., 2680 m a.s.l.) than that of the relict counterparts
(i.e., 2290 m a.s.l.) (**Table 1** and **Fig. 4a**). This altitudinal separation is about constant across aspects, with median front
elevations increasing progressively from northern facing slopes and reaching a maximum on southern aspects (**Fig. 4a**). The
same mismatch is mirrored by the altitudinal distribution of total rock glacier area (**Fig. 4b**), in which intact and relict rock
glaciers are about normally distributed and peak respectively at elevation bands of 2600-2800 and 2200-2400 m a.s.l.

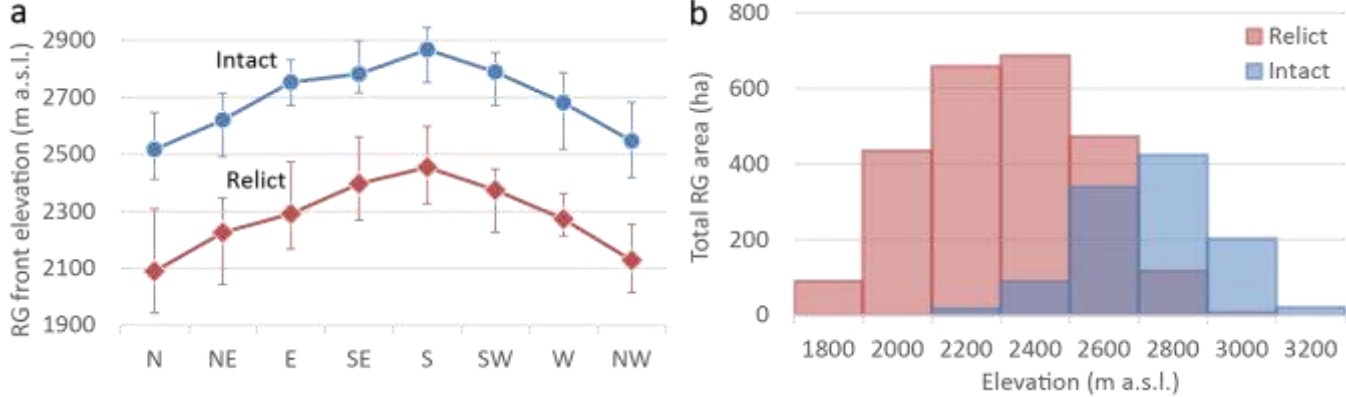

**Figure 4.** (a) Median elevation of rock glacier fronts as a function of slope aspect. (b) Total rock glacier area across
altitudinal bands. In panel a, bars enclose interquartile ranges. In panel b, note altitudinal overlap (i.e., 2200-2800 m a.s.l.)
between intact and relict rock glaciers.

Table 1. Rock glaciers stratified by degree of activity in the geomorphologic inventory, based on interpretation of optical
imagery and some confirmatory fieldwork.

| RG activity | Number of obs. (%) | Median RG front elevation (m a.s.l.) | Median RG size (ha) | Total RG area (ha) |
|---|---|---|---|---|
| Intact | 322 (41) | 2680 | 2.00 | 1090 |
| Relict | 467 (59) | 2290 | 3.16 | 2465 |
| Total | 789 | 2440 | 2.62 | 3555 |





## 4.2 Rock glacier kinematic inventory and integration with the geomorphological approach

Through visual inspection of 2018 and 2019 interferograms (**Fig. 1c** and **3**) we have identified and manually delineated a total of 656 moving areas (MAs) (**Table 2**). Of these, 640 lie within 326 (out of 789) rock glacier polygons previously mapped on optical imagery (**Table 1** and **Fig. 5a**), and 16 belong to 14 newly detected rock glaciers, which were missed
during the compilation of the geomorphologic inventory. Overall, mapped moving areas span across the five velocity classes considered (**Table 2**) and impart to hosting rock glaciers annual velocities comprised between "cm" and "dm to m" (**Table 3** and **Fig. 5b**).

Table 2. Moving areas detected within rock glacier polygons and stratified by velocity.

| MA velocity (cm yr$^{-1}$) | Number of obs. (%) | Median MA elevation (m a.s.l.) | Median MA size (ha) | Total MA area (ha) |
|---|---|---|---|---|
| 1 – 3 | 208 (32) | 2545 | 0.58 | 196 |
| 3 – 10 | 162 (24) | 2695 | 0.59 | 147 |
| 10 – 30 | 168 (26) | 2755 | 0.76 | 190 |
| 30 – 100 | 107 (16) | 2710 | 1.28 | 210 |
| > 100 | 11 (2) | 2650 | 0.66 | 9 |
| Total | 656 | 2680 | 0.73 | 752 |

Based on the spatial distribution of moving areas, we have classified respectively 340 (42%) "moving", 319 (40%) "not moving", and 144 (18%) kinematically "undefined" rock glaciers (**Table 3**, **Fig. 5b** and **Fig. 6**). Where undefined are those landforms for which, due to geometric distortions (layover/shadowing), atmospheric artefacts, phase bias, or decorrelated pattern in the interferograms, surface displacement could not be assuredly determined. Among those bearing movement, 145 (43%) were found to displace at cm annual rate, with the remaining 195 evenly distributed across cm-to-dm, dm, and dm-to-
m kinematic classes (**Table 3**). Interestingly, faster moving areas, and consequently faster rock glaciers, appear to occupy (on average) higher elevations than slower ones, as suggested by their median elevation stratified by velocity class, which peak respectively for moving areas that displace at 10-30 cm yr$^{-1}$ i.e., rock glaciers that move decimetre annual rates (cf., **Tables 2** and **3**).

With reference to the original geomorphologic inventory, InSAR analysis confirmed that 246 (76%) rock glaciers originally
interpreted as intact, do exhibit movement, and that 270 (60%) of the relict labelled counterparts, do not display detectable surface displacement (**Table 4** and **Fig. 6**). At the same time, InSAR contribution proved critical for reclassifying 121 (15%) rock glaciers, clarifying that 41 (13%) of those initially interpreted as intact, do not exhibit detectable movement (i.e., ≥ 1 cm yr$^{-1}$), and that 80 (17%) of the relict ones do actually move (**Table 4** and **Fig. 6**). Among the rock glaciers that have remained kinematically undefined, the proportion of those initially interpreted as relict (23%; n = 109) is much larger than that of the
morphologically intact ones (11%; n = 35), suggesting that the former morphological type is subject to higher uncertainty. As previously noted, in the "integrated" version of the inventory, these landforms will retain their original morphologically-based dynamic classification (**Fig. 6**).




**Figure 5.** Spatial distribution of rock glacier centroids stratified by the dynamic classification of: (a) the geomorphologic approach; (b) the InSAR-based kinematic approach, including annual velocity classes; and (c) the integrated approach and including membership changes and newly-detected landforms (cf., **Fig. 2**). Shaded relief from the Autonomous Province of Bolzano (https://geoportale.retecivica.bz.it/geodati.asp; last access: June 2023).


Table 3. Rock glacier characteristics across InSAR-based kinematic classes. Moving rock glaciers include 14 additional units
identified during multi-temporal examination of interferograms.

| RG kinematic class | Number of obs. (%) | Median RG front elevation (m a.s.l.) | Median RG size (ha) | Total RG area (ha) |
|---|---|---|---|---|
| cm yr$^{-1}$ | 145 (18) | 2475 | 2.63 | 736 |
| cm to dm yr$^{-1}$ | 63 (8) | 2625 | 1.69 | 187 |
| dm yr$^{-1}$ | 68 (8) | 2720 | 2.73 | 329 |
| dm to m yr$^{-1}$ | 64 (8) | 2640 | 3.14 | 287 |
| Moving | 340 (42) | 2585 | 2.61 | 1539 |
| Not moving | 319 (40) | 2360 | 2.52 | 1266 |
| Undefined | 144 (18) | 2100 | 3.06 | 779 |
| Total | 803 (100) | 2440 | 2.62 | 3584 |

Table 4. Rock glacier dynamic classification and membership changes following integration of the geomorphologic and
kinematic inventorying approaches.

| Integrated classification | Number of obs. (%) | Median RG front elevation (m a.s.l.) | Median RG size (ha) | Total RG area (ha) |
|---|---|---|---|---|
| Remain intact | 281 (35) | 2675 | 2.10 | 1014 |
| Become intact | 80 (10) | 2375 | 3.74 | 534 |
| Newly-detected intact | 14 (2) | 2765 | 1.18 | 31 |
| Total intact | 375 (47) | 2610 | 2.40 | 1579 |
| Remain relict | 387 (49) | 2270 | 3.06 | 1931 |
| Become relict | 41 (5) | 2725 | 1.22 | 74 |
| Total relict | 428 (53) | 2300 | 2.82 | 2006 |

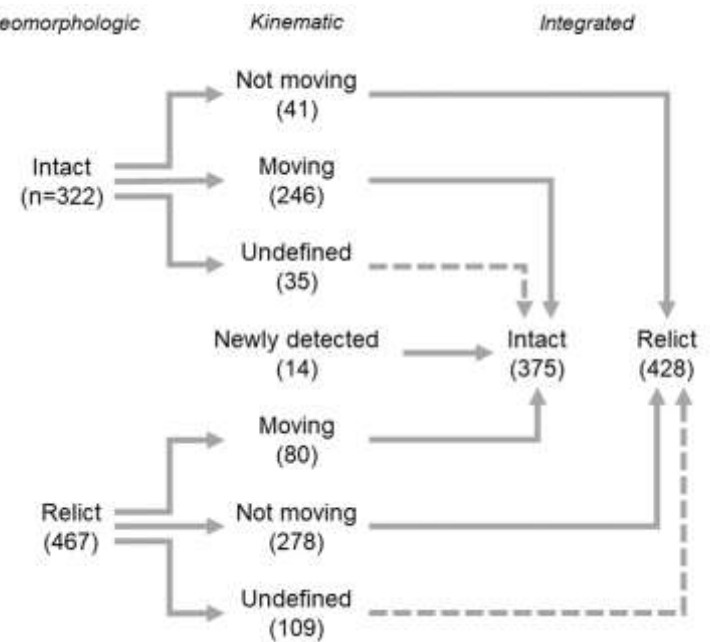





**Figure 6.** Dynamic classification of rock glaciers and relevant membership changes across the geomorphologic, InSAR-based (i.e., kinematic), and integrated inventories. Dashed lines indicate uncertainty in membership attribution associated with undefined cases. See Fig. 2 and relevant text for information on decision rules adopted in the integrated inventory.

**4.3 Topographic characterization of moving areas and distribution within intact rock glaciers**

To fully exploit InSAR-based information and improve our understanding of rock glacier occurrence and dynamics in relation to present topo-climatic conditions, we examine how the velocity and spatial distribution of moving areas within rock glaciers varies as a function of simple topographic variables such as elevation and aspect, known to exert first-order controls on permafrost occurrence. The altitudinal distribution of moving areas displays a progressive increase from north- to south-facing slopes (**Fig. 7**), grossly confirming the strong dependence on slope aspect previously suggested by the

qualitative geomorphologic approach (**Fig. 4a**). This dependence is ill-defined for "slowest" moving areas (i.e., 1-3 cm yr$^{-1}$; **Fig. 7a**), which display high scatter at elevations below 2600 m, and becomes better constrained for 3-10 cm yr$^{-1}$ and 10-30 cm yr$^{-1}$ counterparts (**Fig. 7b** and **7c**). In this context, faster moving areas (i.e., > 30 cm yr$^{-1}$), while broadly following the same pattern across aspects, exhibit a remarkable gap on dominantly south facing slopes (**Fig. 7d**). Overall, the upper altitudinal limit of areas moving at 1-3 cm yr$^{-1}$ plots about 300 m below that of the other (faster) classes, which, by contrast,

do not show altitudinal segregation from each other (**Fig. 7b** through **7d** and **Table 2**).

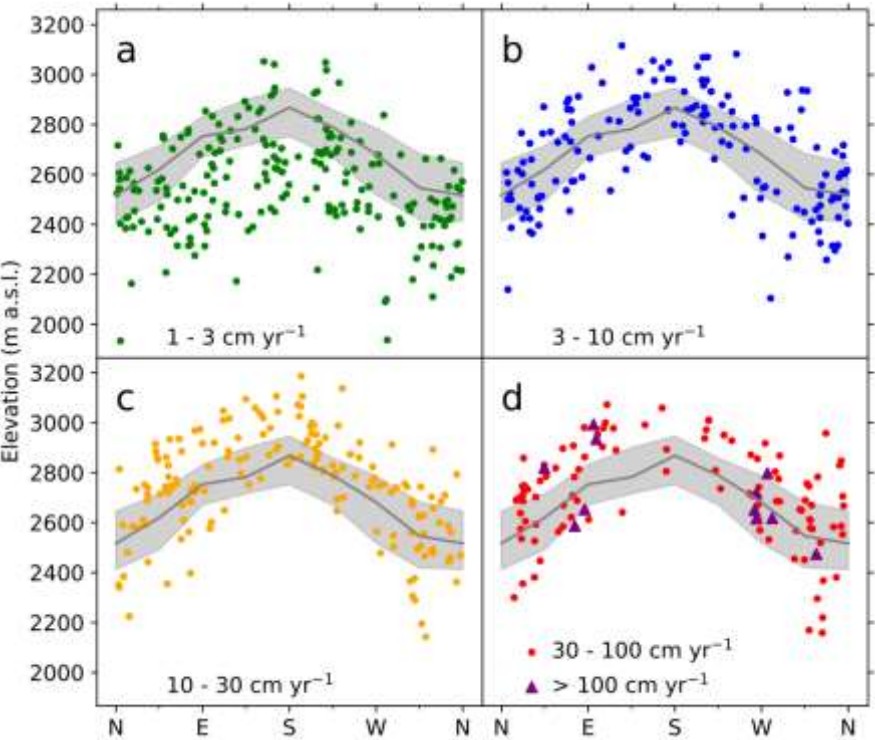

**Figure 7.** Altitudinal distribution of moving areas as a function of slope aspect and stratified by velocity: (a) 1-3 cm yr$^{-1}$; (b) 3-10 cm yr$^{-1}$; (c) 10-30 cm yr$^{-1}$; and (d) 30-100 cm yr$^{-1}$ and >100 cm yr$^{-1}$. Grey shaded area and black linework indicate





respectively the interquartile range ($25^{th}$ to $75^{th}$ percentile) and the median elevation of intact rock glacier fronts as mapped
and classified in the geomorphologic inventory.

Mapped moving areas range in size from 0.08 to 21 ha, with an overall median of 0.73 ha. Their median size, which does not
vary systematically with velocity class, peaks distinctively for those that move at 30-100 cm $yr^{-1}$ (i.e., 1.28 ha; **Table 2**).
When considered against slope aspect, MA size exhibits no obvious dependence. Specifically, we observe a consistent, wide
range of size variability across aspects, with relevant median sizes ranging from as little as 0.57 ha on western aspects up to
0.96 ha on south-facing positions (red triangles in **Fig. 8a**). This pattern does not change when considering moving areas
slower than 30 cm $yr^{-1}$ (**Fig. S1a** through **S1c** in supplementary material), whereas a cluster of small (i.e., area < 0.4 ha)
moving areas stands out on eastern, western and north-western aspects for velocities higher than 30 cm $yr^{-1}$ (**Fig. S1d**). MA
size does not appear to correlate with elevation either. Accordingly, if we exclude moving areas below 2200 m a.s.l. – which,
however, have a limited sample size – consistent high scatter and comparable MA median sizes (red triangles range from
0.57 ha (2000-2200 m a.s.l.) to 0.86 ha (2800-3000 m a.s.l.) are observed across elevations (**Fig. 8b**), also when moving
areas are stratified by velocity (**Fig. S2**).

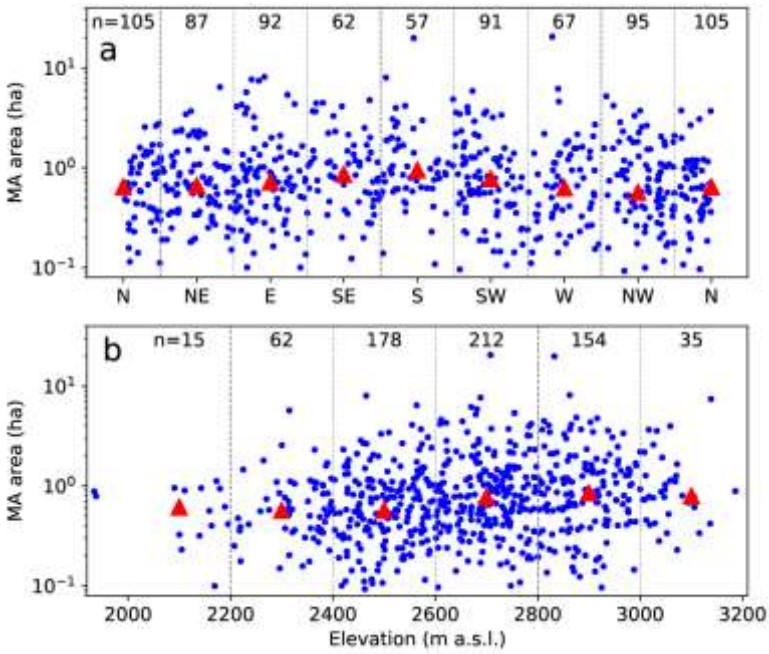

**Figure 8.** Moving area (MA) size represented as a function of: (a) slope aspect; and (b) elevation. Red triangles indicate
median MA values across respectively aspect sectors and altitudinal zones, bounded by dashed lines.

To explore which proportion of a rock glacier polygon (as delineated in the geomorphologic inventory) actually moves and
how this proportion may vary as a function of rock glacier size and elevation, we have represented *total moving area* – here
defined as the combined areal extent of the moving areas hosted within a given rock glacier – against rock glacier size,





across four altitudinal bands (**Fig. 9**) and four velocity classes (**Fig. 10**). Data stratified by elevation show high variability

below 2500 m a.s.l., ranging from cases in which virtually the entire rock glacier surface moves (1:1 line) to cases where the

fractional moving extent drops to less than 10% (1:10 line in **Fig. 9a**). At progressively higher elevations this variability

declines and we observe that total moving area increasingly clusters between 50% (1:2 line) and 100% (1:1 line) (cf. **Fig. 9b**

through **9d**). At elevations above 2900 m a.s.l., where clustering is highest and data trend parallel to isometry (1:1 line) total

moving area increases at about the same rate of rock glacier size.

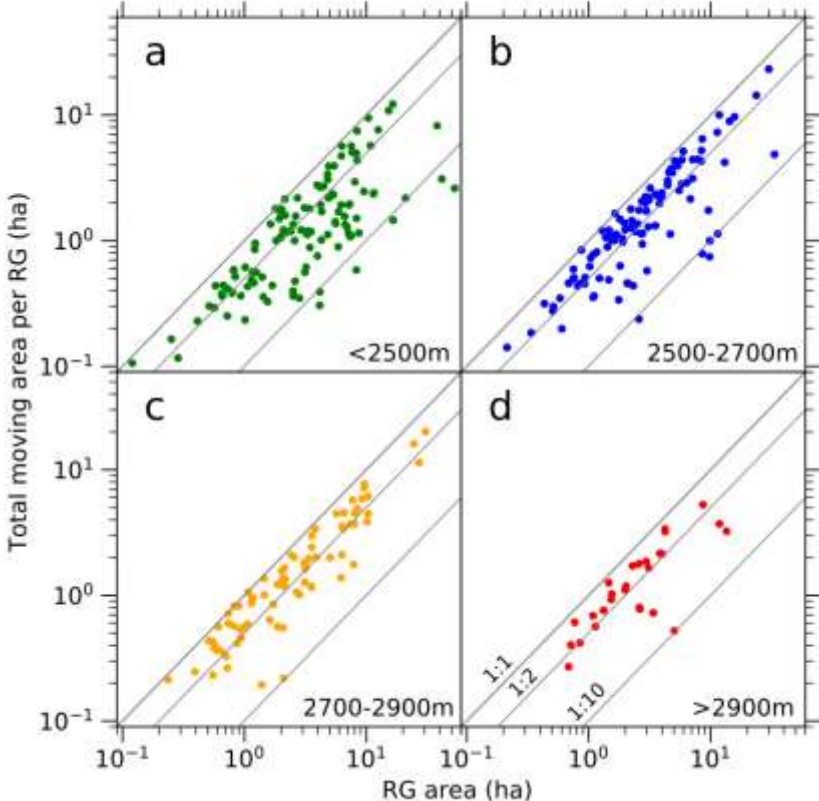

**Figure 9.** Total moving area per rock glacier as a function of rock glacier size across elevation bands: (a) below 2500 m
a.s.l; (b) between 2500 m and 2700 m; (c) between 2700 and 2900 m; and (d) above 2900 m. Solid lines indicate reference
ratios corresponding to percentages of moving area cover within a rock glacier: 1:1 (100% cover), 1:2 (50% cover), and 1:
10 (10% cover).

When rock glaciers are stratified by surficial velocity of displacement along LOS, we observe patterns of variability similar

to those just described across elevation bands. Accordingly, scatter is highest for rock glaciers moving at cm annual rates,

where total moving area ranges from less than 10% to 100% of their surface (i.e., points cluster between 1:10 and 1:1 lines in

**Fig. 10a**), and drops rather abruptly for faster rock glaciers, where at least 30 to 40% of their surface moves (i.e., points

cluster dominantly between 1:2 and 1:1 lines in **Fig. 10b** through **10d**). Collectively, the systematic variability of total




moving area suggests that as rock glaciers move faster, an increasingly larger proportion of their surface becomes kinematically involved, and that this proportion increases with elevation.

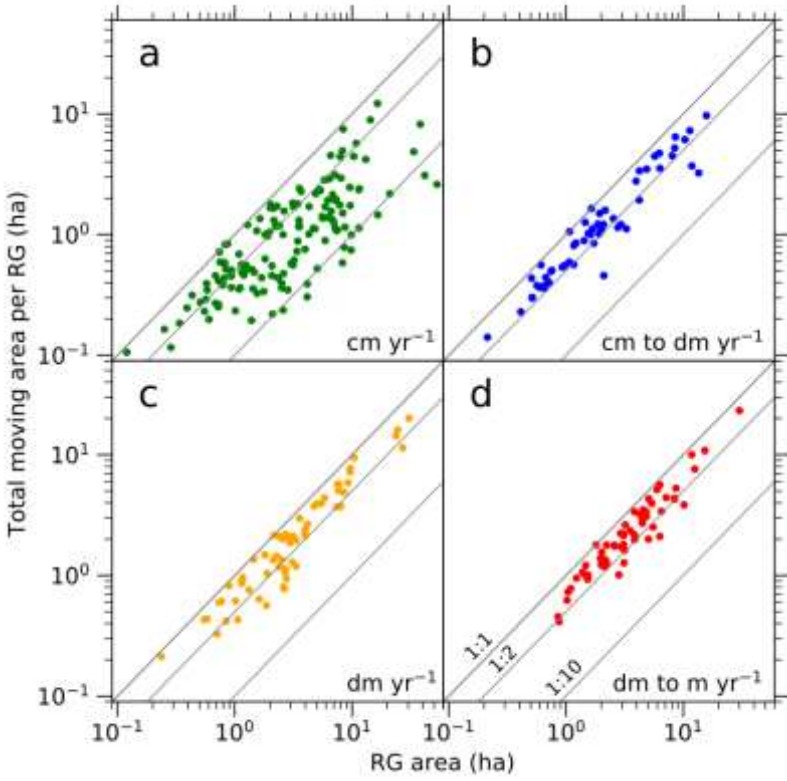

**Figure 10.** Total moving area per rock glacier as a function of rock glacier size stratified by annual rate of surficial displacement: (a) cm yr$^{-1}$; (b) cm to dm yr$^{-1}$; (c) dm yr$^{-1}$; and (d) dm to m yr$^{-1}$. Solid lines indicate reference ratios corresponding to percentages of moving area cover within a rock glacier: 1:1 (100% cover), 1:2 (50% cover), and 1:10 (10% cover).

## 5. Discussion

The compilation and maintenance of a rock glacier inventory, which includes both intact and relict landforms is critical for advancing knowledge on a variety of basic and applied challenges associated with the changing high-mountain cryosphere. For example, the kinematic characterization and monitoring of intact rock glaciers allows evaluating contingent responses to warmer climate conditions, which may range from transition to relict morphologies, with widespread surface subsidence and loss of downslope momentum (Bollmann et al., 2015; Necsoiu et al., 2016), to generalized rock glacier destabilization with orders of magnitude increase in mean annual velocity at their fronts (Delaloye et al., 2008; Scotti et al., 2017a; Marcer et al., 2019; 2021; Dunn et al., 2022) and enhanced potential for debris-flow occurrence (Kummert and Delaloye, 2018; Kummert et al., 2018; Kofler et al., 2021). Similarly, detecting and mapping relict rock glaciers is critical for the appraisal of past





environmental conditions that fostered permafrost development and persistence (e.g., Frauenfelder et al., 2001; Boeckli et
al., 2012; Schmid et al., 2015), as well as for reconstructing geomorphic responses to Holocene climatic changes (e.g.,
Krainer et al., 2015; Zasadni and Kłapyta, 2016; Scotti et al., 2017b; Amschwand et al., 2021).

With respect to the original geomorphologic inventory, we first discuss the changes imparted by InSAR information to the
dynamic classification of rock glaciers, consolidate the relevant gain in inventorying reliability, and acknowledge the
remaining sources of uncertainty (**section 5.1**). Subsequently, starting from a set of representative case studies, we illustrate
local changes associated with the InSAR-based dynamic classification of rock glaciers – from intact to relict, and vice versa
– and illustrate how these translate into valley-wide changes to the altitudinal separation (or overlapping) between relict and
intact landforms across slope aspects (**section 5.2**). Finally, not accounting for variable slope aspect, we identify an
elevation-dependent increase in average rock glacier velocity and average MA cover and discuss its geomorphic significance
with respect to the current state of mountain permafrost.

## 5.1 InSAR-based kinematic information and sources of uncertainty

In this contribution, we illustrate the importance of complementing a geomorphologic rock glacier inventory with InSAR-
derived kinematic information. Integration of the two inventorying approaches, each of which characterized by specific
strengths and weaknesses, allows reducing the overall uncertainty associated with the procedures of rock glacier detection,
mapping and dynamic classification. On one hand, visual interpretation of multi-temporal optical imagery is crucial for
outlining the morphological footprint of a rock glacier, and serves as a benchmark for developing automated mapping
routines (e.g., Robson et al., 2020; Reinosch et al., 2021); on the other hand, visual interpretation of multi-temporal wrapped
interferograms – except where decorrelation occurs – ultimately determines whether a rock glacier exhibits surface
deformation, at what mean annual rate, and in which portion of its morphological footprint, thus confirming or rectifying
prior evaluation solely based on the interpretation of morphological attributes. This InSAR-based dynamic (re)classification,
however, must take into account: (i) the minimum phase difference (thus the minimum rate of annual displacement) that a
given SAR constellation can capture (i.e., ≈ 1 cm yr$^{-1}$ for Sentinel-1 annual interferograms); and (ii) the underestimation of
interferometric phase change due to unfavorable geometry, even when both ascending and descending acquisitions are
examined (Klees and Massonnet, 1998; Liu et al., 2013).

Computation of the α angle between the downslope direction of each rock glacier's main flow line and the LOS (cf.,
equation 1 in **section 3.2.1**) aided identifying in the aspect-slope space (**Fig. 11a**) 177 landforms (out of 789) associated with
phase underestimation that exceeds 50% and therefore deemed unreliable (e.g., Klees and Massonnet, 1998). In agreement
with prior studies (e.g., Liu et al., 2013; Strozzi et al., 2020), we find that similar rates of underestimation primarily affect
rock glaciers located around southerly aspects (n = 131) and secondarily around northerly ones (n = 46) (i.e., circles with red
outline in **Fig. 11a**). Among these, 61 (34%) bear moving areas and as such are classified as "moving", 88 (50%) lack of
moving areas and therefore are regarded as "not moving", and 28 (16%) are labelled as "undefined" due to decorrelation.
While for the "moving" ones high underestimation would mean – in the worst case scenario – assigning these landforms to a





slower (than real) kinematic class, part of the 88 "not moving" rock glaciers may be just the result of signal underestimation, which for example would preclude detection of rock glacier portions moving at cm annual rates, and consequently lead to misclassification (i.e., not moving rather than actually moving).


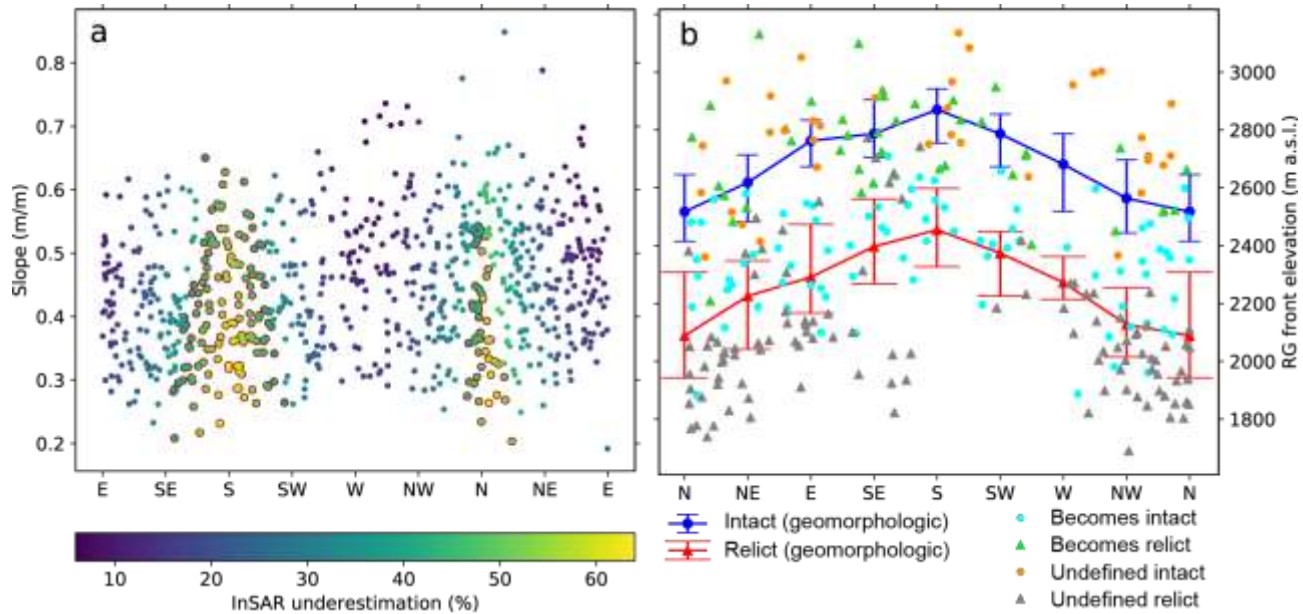

**Figure 11.** (a) Color-coded, percent underestimation of interferometric phase signal as a function of mean values of aspect and slope gradient. For each rock glacier, we consider and represent graphically the lower value of underestimation between the ascending and descending configurations. Red circles identify rock glaciers with underestimation > 50%, which we regard affected by high uncertainty. (b) Scatterplot showing in the aspect-elevation space the position of rock glaciers that underwent dynamic reclassification following InSAR analysis and of those that are considered undefined (due to decorrelation), which therefore will retain the original, geomorphologic-based dynamic classification (i.e., undefined intact and undefined relict). For reference, we report the median and the interquartile range of altitudinal variability (i.e., whiskers) of relict (red) and intact (blue) rock glaciers, as originally classified in the geomorphologic inventory.


A last source of uncertainty is represented by decorrelation, which affects 144 (18%) landforms (**Table 3** and **Fig. 6**). Of these, 109 had been classified as morphologically relict, and 35 as morphologically intact. For the former set, visual examination of optical imagery suggests that decorrelation is likely associated with extensive vegetation cover and therefore during aerial photo interpretation they were assumed to bear no motion (e.g., Barsch, 1992; Haeberli et al., 2006; Scotti et al., 2013). This qualitative interpretation appears reasonable, considering that most of the kinematically-undefined, morphologically-relict rock glaciers plot in the aspect-elevation space at low elevations (grey triangles in **Fig. 11b**). In particular, 88 (out of 98) lie within the interquartile altitudinal range of relict rock glaciers across slope aspects. The same reasoning can be extended to the 46 kinematically-undefined, morphologically-intact rock glaciers, considering that all but four of them lie within the interquartile altitudinal range of intact rock glaciers (orange dots in **Fig. 11b**). In this second set of





rock glaciers, decorrelation is probably chiefly related to issues including persistent snow cover, atmospheric artefacts, phase bias, or extended layover and shadowing (Barboux et al., 2014; Yague-Martinez et al., 2016).

Despite the caveats noted in this section, the dynamic reclassification of rock glaciers based on InSAR kinematic information – for landforms in which signal underestimation falls below 50% – has afforded a notable reduction in the uncertainty of the morphological inventory. This is apparent when considering that 80 rock glaciers previously regarded as relict, did bear

moving areas within their footprints, and that 41 of those previously classified as intact, did not host any (**Fig. 6**). The significance of this InSAR-based reclassification procedure is further supported by the distribution of the relevant rock glaciers in the aspect-elevation space. Accordingly, the vast majority of rock glaciers that have respectively "become intact" (70 out of 80) and "become relict" (39 out of 41) falls within the altitudinal domain (i.e., interquartile range) of the opposite dynamic category (**Fig. 11b**), thus indirectly weakening the altitudinal separation previously observed between

morphologically intact and relict landforms (**Fig. 4a**). In particular, we observe that the median front elevation of rock glaciers that "become intact" (2375 m a.s.l.) approaches that of counterparts that "remain relict" (2270 m) (**Table 4**); vice versa, the median elevation front of those that "become relict" (2725 m) closely matches that of counterparts that "remain intact" (2675 m) (**Table 4**). In the context of uncertainty reduction through integration of different inventorying approaches, InSAR contribution allowed lessening what appears to be a systematic, altitudinal-driven bias that may affect operators'

visual interpretation of landforms. Our findings suggest that operators, to some extent, may base their dynamic classification of rock glaciers – especially when these do not exhibit unambiguous morphological evidence – on the elevation at which they are located.

**5.2 InSAR-based dynamic re-classification of rock glaciers: from local examples to valley-wide effects**

A cluster of six rock glaciers lying on a west-facing slope in Solda Valley (**Fig. 12**) is instructive, both for illustrating how

complex the spatial distribution of moving areas can be, hence for exemplifying the range of possible cases of agreement, disagreement and indetermination associated with the joint evaluation of the morphological and kinematic approaches. Agreement applies to the uppermost intact landform (RG1), in which two moving areas are detected – a primary one (30-100 cm yr$^{-1}$) that covers the entire rock glacier surface, and a secondary faster one (> 100 cm yr$^{-1}$) located in the central portion – as well as to two relict rock glaciers (RG2 and RG3) further downslope, in which no moving areas were detected. Nearby,

roughly at the same elevation, lie kinematically undefined RG4 and RG5, as well as RG6, morphologically classified as relict but later reclassified as intact, in view of the two moving areas detected on interferograms. This reclassification imparts a local drop (of about 100 m) to the lower limit of intact rock glaciers.

A second cluster of landforms on a slope facing south-to-southeast in Martello Valley (**Fig. 13**) exemplifies dynamic classification agreement for intact rock glaciers RG1 and RG2 – in which moving areas span from the rooting zones down to

the relevant fronts – and relict counterparts RG3 and RG4, but underlies disagreement for RG5 and RG6. The former hosts moving areas (3-10 to 10-30 cm yr$^{-1}$) in the rooting zone and across its upper half (which qualifies as an example of "inactive" rock glacier, following the definition by Barsch (1996), see **section 3.1**), the latter exhibits surficial displacement,



even though at the lowest detectable rate (1-3 cm yr$^{-1}$), across the entire surface. As a result, the lower limit of intact rock glaciers drops locally from 2850 m down to 2300 m a.s.l.

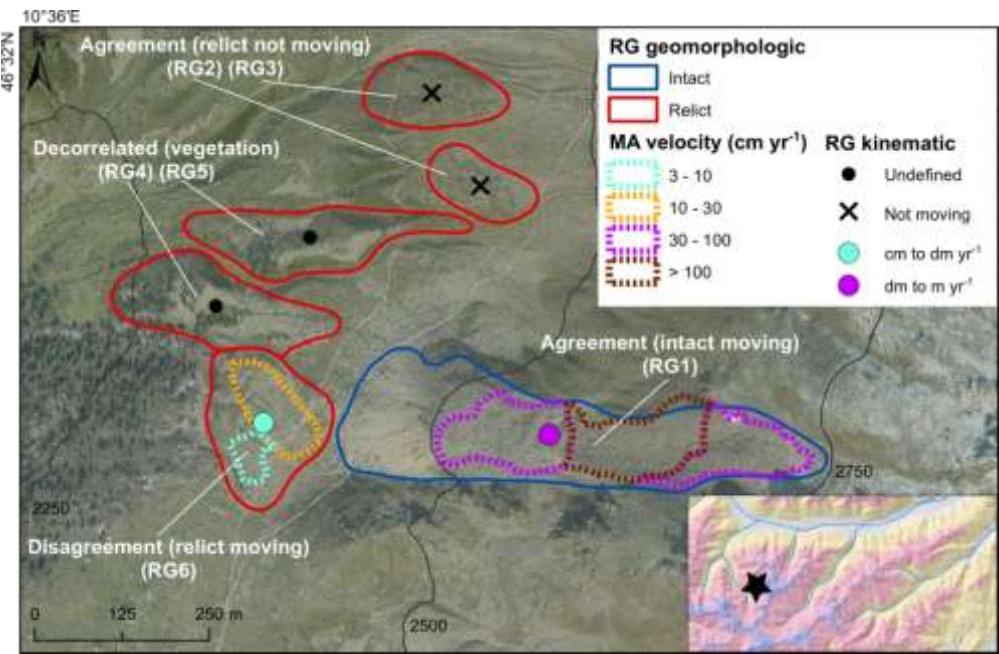

**Figure 12.** Example of six rock glaciers in Solda Valley. The intact rock glacier RG1 contains moving areas and is classified as moving (i.e., at dm to m yr$^{-1}$ rate). At lower altitude, relict rock glaciers RG2 and RG3 lack of moving areas, and other two landforms are affected by InSAR decorrelation due to vegetation (i.e., "Undefined"; RG4 and RG5). RG6 originally classified as relict, hosts moving areas and is reclassified as intact. At this location, SAR underestimation is lower than 20%. Orthoimage from the Autonomous Province of Bolzano (https://geoportale.retecivica.bz.it/geodati.asp; last access: June 2023).

InSAR-based classification can also impart a local rise to the altitudinal distribution of relict rock glaciers. This is the case of RG1 (2585 m a.s.l.) and RG2 (2575 m a.s.l.) in Ultimo Valley (**Fig. S3a**), originally classified as intact, but that do not host any moving areas (i.e., InSAR underestimation < 20%), and therefore in the integrated classification become relict. A last example is drawn from Martelltal/Martello Valley (**Fig. S3b**). At this northerly facing site, despite InSAR underestimation ranging from 39% to 57%, three out of four rock glaciers are found to host moving areas, with velocity that ranges from 3-10 cm yr$^{-1}$ in RG1 (at 2300 m a.s.l., formerly classified as relict) to 30-100 cm yr$^{-1}$ in RG3 (at 2750 m a.s.l.). In this context of high phase underestimation, the lack of moving areas in RG4 should be considered with caution, as slow deformation (i.e., 1-3 cm yr$^{-1}$) might be occurring.

Beyond local effects, integration of InSAR-based kinematics imparts substantial changes to the overall altitudinal distribution of intact and relict rock glaciers, as originally assessed in the geomorphologic inventory (**Fig. 14**). While the general pattern of progressively increasing front elevation, as one moves from northerly rock glacier fronts (the lowest) to southerly counterparts (the highest), is retained; the altitudinal separation between morphologically intact and relict




landforms (i.e., median values are at least 400m apart across aspects) (**Fig. 14a**) gets substantially reduced when considering moving and not moving rock glaciers, which display widespread interquartile overlap (i.e., depending on aspect, median separation drops between 200m and 350m) (**Fig. 14b**). Overlap is highest across southeast through southwest facing rock glaciers, whereas westerly oriented landforms change the least and maintain interquartile separation. These altitudinal changes, which are associated with the InSAR-based reclassification of 121 rock glaciers – and therefore neglect

kinematically-undefined landforms (n = 144) – derive from a decrease in the altitudinal distribution of "moving" rock glaciers (compared to the morphologically intact ones) and a simultaneous increase of the "not moving" rock glaciers (compared to the morphologically relict ones), as previously shown (**Table 4** and **Fig. 11b**).

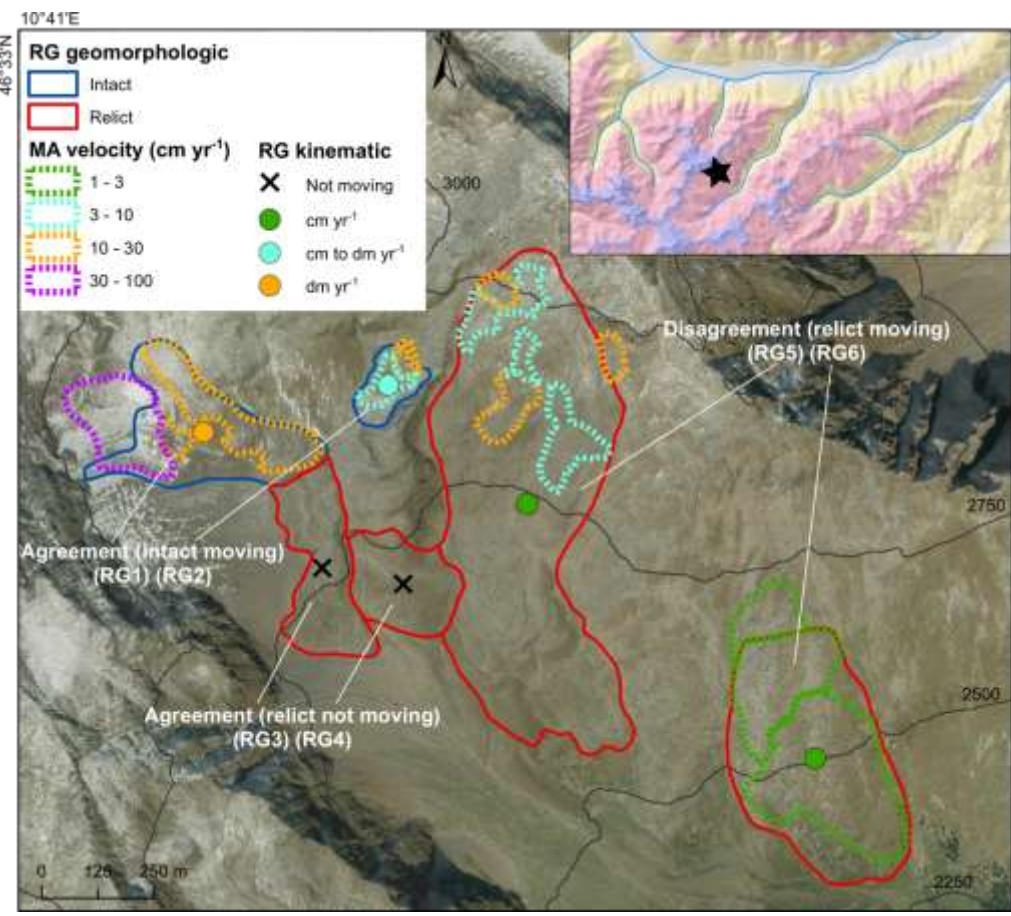

**Figure 13**. Example of six rock glaciers in Martello Valley. Two intact rock glaciers RG1 and RG2, located at high elevation
(front elevation 2775 and 2820 m a.s.l.) host moving areas and are classified as moving. Relict rock glaciers RG3 and RG4 (front elevation 2680 and 2670 m a.s.l.) do not exhibit moving areas and therefore are classified as not moving. However, since underestimation of SAR signal lies close to the reliability threshold (i.e., between 38 and 46%) due to unfavourable geometry with respect to LOS, some degree of uncertainty remains as this configuration could prevent detecting slow surface displacement (i.e., cm yr$^{-1}$). At lower elevations, RG5 (2555 m) and RG6 (2280 m), two relict rock glaciers are indeed
characterized by moving areas: they are classified as moving and consequently become intact in the integrated classification. Orthoimage from the Autonomous Province of Bolzano (https://geoportale.retecivica.bz.it/geodati.asp; last access: June 2023).



When kinematically-undefined landforms are added (i.e., they retain their original morphologically-based membership), interquartile separation between intact and relict landforms in the integrated inventory is partly regained (i.e., median values are 300 m apart) (**Table 4**), except in south-easterly through south-westerly facing landforms, which preserve highest overlap (**Fig. 14c**). In this context, the striking altitudinal separation observed between undefined (morphologically) intact and relict rock glaciers (**Fig. 14d**) suggests that the two clusters may indeed belong to different dynamic categories (i.e.,

moving and not moving) and that our simplistic decision rule of data integration was reasonable.

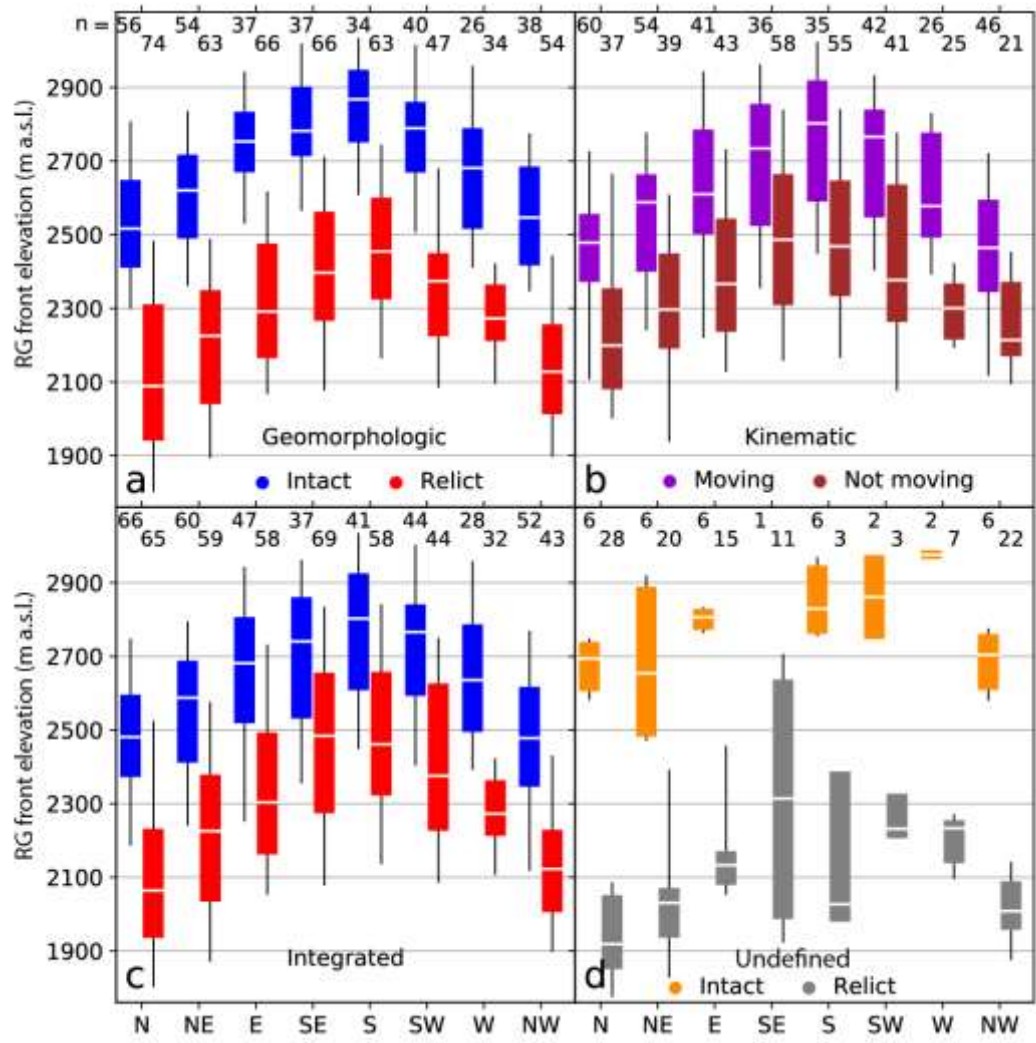

**Figure 14.** Boxplots showing the altitudinal distribution of rock glacier fronts across slope aspects for: (a) intact and relict rock glaciers of the geomorphologic inventory; (b) moving and not moving rock glaciers following InSAR-based kinematic characterization; (c) intact and relict rock glaciers of the integrated inventory; and (d) kinematically-undefined rock glaciers





stratified according to the original morphologically-based, dynamic classification. White horizontal linework represents median values; boxes enclose interquartile range (25th-75th percentiles); and whiskers bound 5th and 95th percentiles.

Overall, the InSAR-based kinematic characterization of rock glaciers, by reducing the altitudinal separation between morphologically intact and relict landforms (**Fig. 14**), depicts a broad transition belt of intact-relict coexistence in the aspect-elevation space, the amplitude of which – here defined as the zone comprised between the 95th percentile of relict rock glaciers and the 5th percentile of intact counterparts – varies from as little as 50 m on west facing slopes to a maximum of 500 m in easterly ones (**Fig. 14c**). A similar altitudinal overlap, which possibly reflects the complex spatial pattern of discontinuous permafrost, deteriorates the significance of elevation and aspect as topographic proxies for modelling permafrost occurrence in south-western South Tyrol, and consequently underlines the importance of using InSAR technology for informing such models.

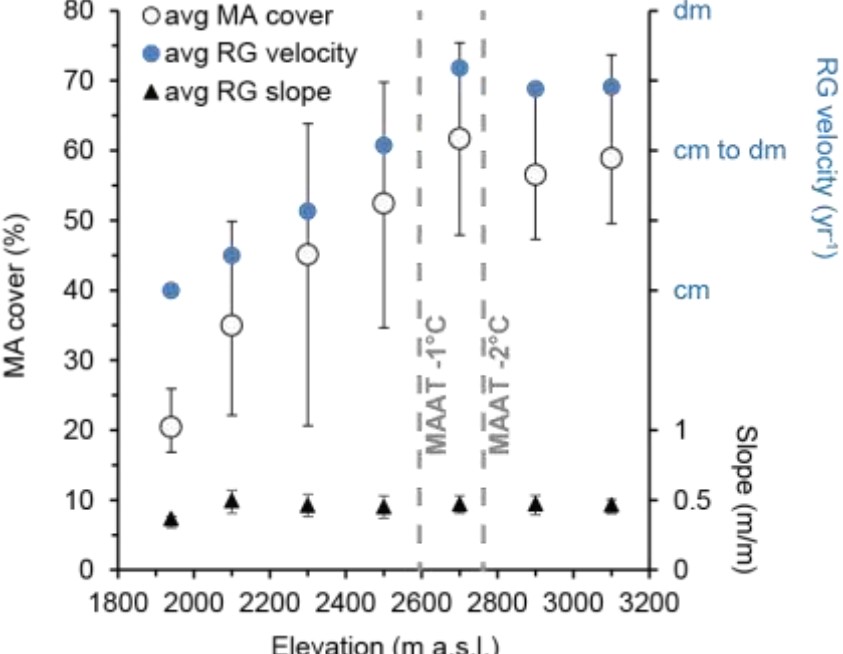

**Figure 15.** Average variation of rock glacier velocity (class), rock glacier surface moving area (MA) cover, and slope gradient (m/m) as a function of elevation. Averages are calculated across the following elevation bands: 1880-2000; 2000-2200; 2200-2400; 2400-2600; 2600-2800; 2800-3000; 3000-3140m. Whiskers around average values bound interquartile ranges (25th-75th percentiles). Velocity classes include: cm yr$^{-1}$; cm to dm yr$^{-1}$, and dm yr$^{-1}$. Note inflection point in velocity and MA cover at elevations comprised between MAAT -1°C and -2°C. MAAT values are computed from gridded data drawn from www.alpenklima.eu and refer to the period 1981-2010.

Equally important, from a basic process-oriented standpoint, InSAR information proves fundamental for imaging how this altitudinal transition manifests through changing rates and styles of rock glacier surface deformation. Collectively, the systematic variability of total moving area suggests that as rock glaciers move faster, an increasingly larger proportion of their surface gets kinematically involved, and that this proportion increases with elevation (**Figures 9** and **10**). This



interpretation becomes apparent when lumping MA variability into rock glacier units, that is, when representing rock glacier annual velocity and percent MA cover (i.e., the proportion of a rock glacier surface occupied by moving areas) as a function of elevation (**Fig. 15**). Indeed, both variables increase linearly with elevation up to the 2600-2800 m band, beyond which an inflection occurs and consistent average values (and ranges of variation) are attained. According to the classification scheme recently proposed by RGIK (2022), the average dynamic state that characterize rock glaciers below and above 2600 m are consistent respectively with *transitional* and *active* rock glacier types.

Our findings differ from similar work conducted in the Uinta Mountains (Utah), in which no apparent altitudinal dependence of rock glacier velocity was found (Brencher et al., 2021). Most importantly, the altitudinal inflection falls between the -1°C (2595 m) and -2 °C MAAT (2760 m) – regarded as the lower boundary for discontinuous permafrost occurrence (Haeberli, 1983; Haeberli et al., 1989) – and appears to be decoupled from local topographic conditions, since average rock glacier slope gradient stays about constant (i.e., around 0.45 m/m) across the entire elevation range (**Fig. 15**). Based on these observations, we argue that the altitudinal increase in average rock glacier velocity and average MA cover represents the dynamic expression of increasing permafrost distribution (i.e., a transition from sporadic to discontinuous), until optimal thermal conditions are reached above 2600 m. To some extent, the increasing activity of rock glaciers with elevation may also have to do with increasing erosion rates on the overhanging rock walls (Draebing et al., 2022), which would supply to the rooting zones increasing amount of debris as frost cracking becomes more efficient. With data at hand, however, we are unable to disentangle permafrost from frost cracking effects.

## 6. Conclusions

In this paper, we show that the integration of InSAR-based information can improve the reliability of a geomorphologic inventory both in terms of completeness, through detection of additional intact rock glaciers on wrapped interferograms, and in terms of dynamic classification, through uncertainty reduction. If on one hand the contribution of the newly-detected rock glaciers is limited, as they account for less than 2%; on the other hand, the dynamic membership is either confirmed or reclassified for respectively 67 % and 15 % of the originally mapped rock glaciers, leaving the remaining 18 % undefined, due to decorrelation. This proportion of undefined cases, which may look high at first glance, drops substantially when considering that the vast majority of these landforms (i.e., 109 out of 144) has been labelled as morphologically relict due to widespread vegetation cover, and therefore, most likely represent "not moving" features. In this context, future work in south-western South Tyrol will aim at reducing the number of undefined rock glaciers by considering longer baselines (e.g., 2 to 3 years) in Sentinel-1, and integrate constellations characterized by different LOS geometry and SAR band (e.g., CosmoSkyMed). This combination would warrant greater flexibility at covering all slope aspects, while lowering the 1 cm $yr^{-1}$ threshold of (assuredly) detectable deformation.

We find that InSAR-based dynamic reclassification of rock glaciers induces substantial increase in the altitudinal overlap between relict and intact rock glaciers across slope aspects. As a consequence, our results portray a more complex picture than that solely based on geomorphologic mapping and indicate that using elevation and aspect as topographic proxies for

discriminating permafrost-bearing from -devoid rock glaciers may be problematic, especially on south-westerly through south-easterly aspects, where overlap is greater. The existence of a broad altitudinal transition zone, in which intact and relict rock glaciers coexist, reinforces the need of using InSAR technology for the site-specific characterization of rock glacier displacement across large spatial scales, hence for inferring the spatial distribution of discontinuous-to-sporadic permafrost
in mountain terrain.

In this context, integration of the two inventorying approaches gains additional importance from a basic, process-oriented standpoint, as it proves particularly effective for imaging how creeping of perennially frozen, ice-debris mixtures varies (in intensity and spatial distribution) as a function of elevation. Specifically, we find that average annual rates of rock glacier surficial displacement and percent MA (moving area) cover – where the latter variable relies both on the morphological
delineation of rock glacier footprints in optical imagery and the kinematic characterization of hosted moving areas in wrapped interferograms – increase linearly with elevation until a sharp inflection occurs and consistent average values are attained above 2600 m a.s.l. Considering that this altitudinal inflection occurs right between the -1°C and -2 °C MAAT – regarded as the lower boundary for discontinuous permafrost occurrence – we interpret this altitudinal pattern as the geomorphic signature of increasing permafrost distribution (i.e., transition from sporadic to discontinuous), until optimal
thermal conditions, which promote "full scale" viscous creep and shearing at depth, are met. Following this logic, we propose that average rock glacier velocity, in conjunction with percent MA cover, be considered as possible metrics for defining the altitudinal transition in rock glacier kinematics and their styles of deformation within and across mountain ranges. This "landscape scale" approach would complement site-scale Essential Climate Variables (ECV) (https://gcos.wmo.int/en/essential-climate-variables), such as rock glacier velocity, that are currently being used to evaluate
sensitivity to climate warming and envisage future scenarios of change.

**Acknowledgements**

This work was funded by the European Space Agency (ESA) Climate Change Initiative (CCI) project [grant number 4000123681/18/I-NB]. AB was supported through a postdoctoral fellowship jointly awarded by the Autonomous Province of Bolzano and the University of Bologna. We thank Gabriel Pellegrinon for aiding with the compilation of the moving area
inventory. We are indebted to colleagues at the University of Fribourg for initiating and managing the IPA Working Group Rock Glacier inventories and kinematics, a great effort that has been instrumental for developing this work.

*Supplement.* The Supplement includes the representation of moving area size as a function of slope aspect (Fig. S1) and elevation (Fig. S2) across kinematic classes; and examples or InSAR-based dynamic reclassification of rock glaciers (Fig.
S3).

*Author contributions.* FB and TS designed the study and managed the project. NJ and TS produced the interferometric data. RS and FB updated and revised the geomorphologic inventory. AB compiled the InSAR-based inventory of moving areas and conducted the kinematic characterization of rock glaciers. AB and FB conducted the statistical analysis on moving areas



and rock glaciers. AB and FB wrote the manuscript. RS and VM contributed to interpretation of the results. All authors
contributed to the drafting of the final version of the paper.

*Competing interests.* The authors declare no competing interests.

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
