# Peer review of "A climate-driven, altitudinal transition in rock glacier dynamics detected through integration of geomorphological mapping and InSAR-based kinematics"

_The Cryosphere, 2023_

## Author Comment (AC1)

**Replies to Referee 1:**

Our replies are embedded in the referee's comments.

We wish to thank the referee for the detailed comments. As a general premise, we would like to clarify that the submitted manuscript relies on the methodological foundations developed by the IPA (International Permafrost Association) Working Group on Rock Glacier Inventories and Kinematics (RGIK) and defined, justified and carefully described in Bertone et al., (2022) and in RGIK (2023). Consequently, in this manuscript we have reported a concise (but complete) description of the methodology involved in the InSAR-based characterization of rock glaciers. Considering the already composite structure and the relevant length of the manuscript, we believe a more detailed description/justification of the InSAR-based methodology is beyond the scope of this paper, as it would make it bulkier and difficult to read by the broader audience of the journal.

Bertone, A, Barboux, C, Bodin, X, Bolch, T, Brardinoni, F, Caduff, R, Christiansen, H H, Darrow, M, Delaloye, R, Etzelmüller, B, Humlum, O, Lambiel, C, Lilleøren, K S, Mair, V, Pellegrinon, G, Rouyet, L, Ruiz, L, and Strozzi, T. 2022. Incorporating InSAR kinematics into rock glacier inventories: insights from eleven regions worldwide. The Cryosphere, 16, 2769-2792. https://doi.org/10.5194/tc-16-2769-2022.

RGIK, 2023. Guidelines for inventorying rock glaciers: baseline and practical concepts (version 1.0). IPA Action Group Rock glacier inventories and kinematics, 25 pp, doi:10.51363/unifr.srr.2023.002

**Comments by Referee 1:**

Thank you for the opportunity to review your paper title "A climate-driven, altitudinal transition in rock glacier dynamics detected through integration of geomorphological mapping and InSAR-based kinematics." The study focuses on the use of InSAR-based information to improve the detection and dynamic classification of rock glaciers in the dry southwestern South Tyrol, Italy. Traditional geomorphologic mapping faces challenges in accurately classifying intact and relict rock glaciers. The research combines InSAR data with geomorphologic approaches to create a more reliable inventory. The findings highlight the significance of InSAR in identifying previously undetected rock glaciers, reclassifying their status, and providing insights into the altitudinal transition of permafrost distribution. The study also emphasizes the importance of InSAR for modelling permafrost occurrence and understanding the dynamics of rock glacier surface deformation in response to changing rates and styles of movement. Overall, I find the study to be valuable, addressing the challenges in accurately classifying rock glaciers and emphasizing the importance of InSAR technology in enhancing inventory reliability.

I have some specific comments that, if incorporated, could further strengthen the study, and contribute to its successful publication.

Abstract:

To enhance clarity and conciseness, the abstract should be streamlined while preserving key objectives and novel contributions. Organize it with a clear structure by introducing the problem, detailing methods, presenting findings, drawing conclusions, and outlining implications. Explicitly highlight the study's significant contributions, ensuring each sentence is concise and contributes to the overall coherence of the abstract.

We would like to thank the referee for the comment. Following this advice, we have cut unnecessary words and details, where deemed desirable (i.e., strikethrough text), and have added a brief sentence on the identification of low-lying intact rock glaciers (i.e., in red), following referee's 2 comment. In this context,

we would value and welcome any specific change that the referee could suggest for making the abstract shorter.

Indeed, we believe that the current abstract meets the criteria kindly reminded by the referee. In particular, the first four sentences are introducing the problem. The following two sentences are detailing the objectives and the methods. Then, the results are presented in the following five sentences. Finally, conclusions are drawn, and implications are outlined in the remaining four sentences of the abstract.

There follows the modified version of the abstract:

"In dry southwestern South Tyrol, Italy, rock glaciers are dominant landforms of the high-mountain cryosphere. Their spatial distribution and degree of activity hold critical information on the  current state of discontinuous permafrost, and consequently on response potential to climate warming. Traditional geomorphologic mapping, however, owing to the qualitative expert-based nature, typically displays a high degree of uncertainty and variability among operators with respect to the dynamic classification of intact (permafrost bearing) and relict (permafrost devoid) rock glaciers. This limits the reliability of geomorphologic rock glacier inventories for basic and applied purposes. To address this limitation: (i) we conduct a systematic evaluation of the improvements that InSAR  can afford to the detection and dynamic classification of rock glaciers; and (ii) build an integrated inventory that  combine**s** the strengths of geomorphologic- and InSAR-based approaches. To exploit fully InSAR-based information towards a better understanding of the topo-climatic conditions that sustain creeping permafrost, we further explore how velocity and the spatial distribution of moving areas (MAs) within rock glaciers may vary as a function of simple topographic variables known to exert first-order controls on incoming solar radiation, such as elevation and aspect. Starting from  a geomorphologic inventory (n = 789), we characterize the kinematics of InSAR-based MAs and the relevant hosting rock glaciers on thirty-six Sentinel-1 interferograms  in the 2018-19 period. With respect to the original inventory, InSAR analysis allowed identifying 14 previously undetected rock glaciers. Further, it confirmed that 246 (76%) landforms, originally interpreted as intact, do exhibit detectable movement (i.e., ≥1 cm yr-1), and that 270 (60%) of the relict labelled counterparts do not, whereas 144 (18 %) resulted kinematically undefined due to decorrelation. Most importantly, InSAR proved critical for reclassifying 121 (15%) rock glaciers, clarifying that 41 (13%) of those interpreted as intact, do not exhibit detectable movement, and that 80 (17%) of the original relict ones do  move. Reclassification, **on one hand allowed identifying a cluster of intact rock glaciers below 2000 m a.s.l., associated with positive MAAT values;** on the other hand, by increasing the altitudinal overlap between intact and relict landforms depicts a broad transition belt in the aspect-elevation space,  which varies from  50 m on west facing slopes to  500 m on easterly ones. This finding deteriorates the significance of elevation and aspect as topographic proxies for modelling permafrost occurrence and highlights the importance of using InSAR for informing such models. From a process-oriented standpoint, InSAR information proves fundamental for imaging how this altitudinal transition manifests through changing rates and styles of rock glacier surface deformation.  We find that as rock glaciers move faster, an increasingly larger proportion of their surface becomes kinematically involved (i.e., percent MA cover), and that this proportion increases with elevation up to the 2600-2800 m, beyond which an inflection occurs, and consistent average values are attained. Considering that the inflection falls between the -1°C and -2 °C MAAT – the lower boundary for discontinuous permafrost – and is independent of slope gradient, we conclude that this altitudinal pattern represents a geomorphic signature: the dynamic expression of increasing permafrost distribution,  from sporadic to discontinuous."

Introduction:

Incorporate a more comprehensive background on the significance of rock glaciers within the context of permafrost and climate change. Refer to relevant recent literature such as Karjalainen et al., 2020, Pruessner et al., 2022, to strengthen the introduction.

Thank you for this comment. Following the reviewer's suggestion we have added reference to Pruessner et al (2022).

Pruessner, L., Huss, M., and Farinotti, D.: Temperature evolution and runoff contribution of three rock glaciers in Switzerland under future climate forcing, Permafrost and Periglacial Processes, 33, 310-322, https://doi.org/10.1002/ppp.2149, 2022.

As for Karjalainen et al (2020), since it deals with the Arctic and circumpolar regions, it does not seem to fit our paper introduction concerned with high-mountain environments and the alpine cryosphere.

We have also modified the first two opening sentences (lines 43-45), making explicit that the references therein refer to the geomorphological and hydrological significance of rock glaciers in a changing climate, while adding reference to ongoing acceleration trends in rock glacier velocity (e.g., Pellet et al., 2023; Kellerer-Pirklbauer et al., 2024), as well as to Li et al (2024) in terms of rock glacier hydrological significance.

Pellet, C., Bodin, X., Cusicanqui, D., Delaloye, R., Kääb, A., Kaufmann, V. Noetzli, J., Thibert, E., Vivero, S. and Kellerer-Pirklbauer, A. 2023. Rock Glacier Velocity. In Bull. Amer. Meteor. Soc. Vol104(9), State of Climate 2022, pp 41-42, doi:10.1175/2023BAMSStateoftheClimate.1

Kellerer-Pirkbauer, A, et al., 2024. Acceleration and interannual variability of creep rates in mountain permafrost landforms (rock glacier velocities) in the European Alps in 1995–2022. Environ. Res. Lett., 19, 034022. https://doi.org/10.1088/1748-9326/ad25a4.

Li, M., Yang, Y., Peng, Z., and Liu, G.: Assessment of rock glaciers and their water storage in Guokalariju, Tibetan Plateau, The Cryosphere, 18, 1–16, https://doi.org/10.5194/tc-18-1-2024, 2024.

The modified sentence now reads as follows (lines 43 and following):

"In the last four decades, atmospheric temperature rise has led to rapid glacial retreat and permafrost degradation in high mountain environments. This trend has promoted slope instability (e.g., Huggel et al., 2015; Kos et al., 2016; Frattini et al., 2016; Schlögel et al., 2020), reduction in rock glacier water storage potential (e.g., Azocar and Brenning, 2010; Jones et al., 2018; Li et al., 2024), and systematic acceleration in rock glacier velocity (e.g., Pellet et al., 2023; Kellerer-Pirklbauer, 2024)."

Introduce a structured guide in the introduction to lead readers through the paper's organization, highlighting key components to be discussed. This will provide a stronger foundation for comprehension.

We are not sure that adding a structured guide at the end of the introduction will help. Considering the current length of the manuscript, and the existing structured guides to the Results (lines 274-278) and to the Discussion (lines 412-419), we are afraid that by duplicating this type of information, the introduction may become redundant and too heavy in the context of the whole paper.

We will leave this decision to the Editor. Please let us know how you would like us to proceed.

There follow two paragraphs meant as guided structure of the paper, that eventually will be added at the end of the introduction (section 1), while removing relevant introductory lines in section 4 and section 5:

"In the presentation of the results (section 4), we start with the dynamic and topographic characterization of the geomorphologic rock glacier inventory. We continue with the InSAR-based kinematic characterization of moving areas and hosting rock glaciers, then we compare the geomorphologic and InSAR-based dynamic classification approaches and combine them in a so-called "integrated" inventory.

We finally examine the velocity and spatial distribution of moving areas in relation to elevation, aspect, and size of the hosting (geomorphologic) rock glacier polygon.

In the discussion (section 5), after introducing several implications and applications tightly linked to the compilation and maintenance of a rock glacier inventory, we first elaborate on the changes imparted by InSAR information to the dynamic classification of rock glaciers in the original geomorphologic inventory. Thus, we consolidate the relevant gain in inventorying reliability, and acknowledge the remaining sources of uncertainty. Subsequently, starting from a set of local examples, we show how local changes associated with the InSAR-based dynamic classification of rock glaciers – from intact to relict, and vice versa – reflect at the valley scale in the altitudinal separation (or overlapping) between relict and intact landforms across slope aspects. Finally, we constrain an elevation-dependent increase in average rock glacier velocity and average MA cover and discuss its geomorphic significance with respect to the current state of mountain permafrost."

Clearly state the study's objectives in the introduction. Articulate the problem the research aims to address, providing a robust rationale for the study.

Thank you for this comment. We believe that the objectives are clearly stated in the introduction, following a statement of the scientific gap to be filled. Accordingly, we enunciate the statement of the problem in lines 81-85:

"Although the application of InSAR technology to rock glacier inventories holds straight forward advantages, amply demonstrated for single, a cluster, or many rock glaciers (e.g., Barboux et al., 2014; Strozzi et al., 2020; Lambiel et al., 2023; Bertone et al., 2023), a systematic and quantitative evaluation of the improvements afforded to a traditional "geomorphologic" inventory, encompassing both intact and relict landforms over broad spatial scales, is missing."

We continue with a paragraph stating how and in which setting we propose to fill this gap (lines 86-89): "To address this gap, following operational guidelines on the InSAR-based kinematic characterization of rock glaciers jointly proposed by ESA Permafrost CCI and IPA Action Group on rock glacier inventories (Bertone et al., 2022), we wish to integrate geomorphologic and InSAR-based inventorying approaches in selected valleys of western South Tyrol, where rock glacier occurrence is overwhelming."

And finally, we state the three main objectives of the paper (lines 89-95):

"Therein, starting from the compilation of a geomorphologic inventory, we aim to: **FIRST OBJECTIVE→** (i) characterize the kinematics of InSAR-based moving areas and relevant hosting rock glaciers; and **SECOND OBJECTIVE→** (ii) evaluate InSAR-derived improvements, in terms of inventory completeness and uncertainty reduction, in the detection and dynamic classification of rock glaciers. To fully exploit InSAR-based information towards a better understanding of the current topo-climatic conditions associated with creeping permafrost, **THIRD OBJECTIVE→** we further aim to explore in which way the velocity and spatial distribution of moving areas within rock glaciers may vary as a function of simple topographic variables known to exert first-order controls on incoming solar radiation and ground temperature, such as elevation and aspect."

Study Area

The description of the study area is comprehensive and provides essential details regarding the rugged mountain terrain in the north-eastern portion of the Ortles-Cevedale massif. The inclusion of elevation ranges, key valleys, and bedrock geology contributes to a clear understanding of the study context.

The information on climate and permafrost occurrence is valuable. However, it will enhance the reader's understanding if you briefly mention the limitations or uncertainties associated with the climate and permafrost data, especially if there are variations in these parameters.

Please consider that this section is a plain description of the study area, meant to provide a concise and broad appraisal of the physiographic boundary conditions.

Precipitation data refer to meteorological stations at easy access sites maintained by dedicated personnel of the Autonomous Province of Bolzano. As such, similar are affected by the same uncertainty that characterizes this type of stations worldwide.

Spatially-distributed data on MAAT (grid 0.5 x 0.5 km), which covers the conterminous regions of Tyrol (Austria), South Tyrol (Italy), and Veneto (Italy), derive from 3PClim (http://www.3pclim.eu/), an Austria-Italy Interreg initiative, funded by the EU. The website is freely accessible to the public in English, in German and in Italian.

The following specifics were added in lines 114-117:

"…, which in southwestern South Tyrol, based on regional climatic characterization (1981-2010), sets between 2595 m and 2765 m a.s.l. (http://www.3pclim.eu/). MAAT gridded data, available at 0.5 km x 0.5 km resolution, were obtained through spatial interpolation of 1460 meteorological stations (Frei, 2014; Hiebl and Frei, 2016), following ad hoc data homogenization (Nemerc et al., 2013). "

Frei, C., 2014. Interpolation of temperature in a mountainous region using nonlinear profiles and non-Euclidean distances. Int J Climatol 34, 1585–1605, doi:10.1002/joc.3786.

Hiebl, J., and Frei, C., 2016. Daily temperature grids for Austria since 1961 – concept, creation and applicability. Theor Appl Climatol., 124, 161-178, doi:10.1007/s00704-015-1411-4.

Nemec, J., Gruber, C., Chimani, B., Auer, I., 2013. Trends in extreme temperature indices in Austria based on a new homogenised dataset of daily minimum and maximum temperature series. International Journal of Climatology 33/6, 1538–1550, doi:10.1002/joc.3532.

PermaNET (www.permanet.eu) is an Alpine Space Interreg initiative (2010-2012), funded by the EU, that for the first time produced orogen-scale Alpine permafrost index map (APIM) across the European Alps. Both concise and extended descriptions of the methodology and outputs may be retrieved respectively in the synthesis report (https://www.permanet-alpinespace.eu/archive/pdf/PermaNET synthesisreport.pdf), and in Boeckli et al. (2012), included in the reference list of our manuscript.

Considering that our work aims to provide improved (i.e., InSAR-based) information for better informing models of permafrost distribution, we believe that adding more information on the PermaNET index map will load the reader with information that, indeed, we do not use/analyze in this paper. We hope that the referee could share our opinion on this.

For convenience, there follows the Abstract by Boeckli et al (2012):

Abstract. The objective of this study is the production of an Alpine Permafrost Index Map (APIM) covering the entire European Alps. A unified statistical model that is based on Alpine-wide permafrost observations is used for debris and bedrock surfaces across the entire Alps. The explanatory variables of the model are mean annual air temperatures, potential incoming solar radiation and precipitation. Offset terms were applied to make model predictions for topographic and geomorphic conditions that differ from the terrain features used for model fitting. These offsets are based on literature review and involve some degree of subjective choice during model building. The assessment of the APIM is challenging because limited

independent test data are available for comparison and these observations represent point information in a spatially highly variable topography. The APIM provides an index that describes the spatial distribution of permafrost and comes together with an interpretation key that helps to assess map uncertainties and to relate map contents to their actual expression in the terrain. The map can be used as a first resource to estimate permafrost conditions at any given location in the European Alps in a variety of contexts such as research and spatial planning.

Also, if the the authors can address potential challenges related to the chosen boundaries, terrain characteristics, or data constraints that might influence the study outcome.

The study area was selected on purpose within the Western Austroalpine physiographic region of South Tyrol to warrant consistent boundary conditions (Scotti et al., in press). In particular, the area encompasses alpine tributary valleys sharing the same Quaternary (glacial and postglacial) landscape history, as well as homogeneous lithologic, topographic and climatic conditions.

Data collection and analysis

3.1

Authors please explicitly mention the sources of the optical imagery (2014, 2017, 2020) and LiDAR-derived hill shade raster (2006). This clarification will provide readers with a clear understanding of the data utilized.

Web links to arial photo-imagery and LiDAR-derived hillshade raster were provided in the captions of Figures 3, 5, 12 and 13. To address the reviewer's concern we have added such information in section 3.1. Accordingly, the revised sentence in lines 125-128 now reads as follows: "The identification, mapping and dynamic classification of each rock glacier relies primarily on the visual interpretation of 0.2-m gridded, optical imagery (i.e., orthophoto mosaics flown in 2014, 2017, and 2020) and a 2.5-m LiDAR-derived hillshade raster (i.e., 2006) -- all available as WMS resources at the Geological Web Portal of the Autonomous Province of Bolzano (https://geoportale.retecivica.bz.it/geodati.asp) -- and where available, on information drawn from local reports on road closure or damage to infrastructure associated with the advance of rock glacier fronts."

It will be preferable while the explanation of dynamic classification in detailed, consider summarizing it briefly or using a bulleted list for easy reference.

The dynamic classification of rock glaciers into three classes occupies less than six lines of text (lines 134-138). We are afraid we cannot summarize it (condense it) any further without losing key information for the broader readership of the journal. We adopt the widely accepted, international classification scheme put forward by Haeberli (1985) and Barsch (1996), which form classical keystones in rock glacier and permafrost literature.

When referring to previous studies (e.g., Brardinoni et al., 2019; Haeberli, 1985), consider providing brief context about the study to help readers access additional information.

Context to Brardinoni et al (2019) has been provided in lines 60-65 of the Introduction. Being a classical milestone in rock glacier studies, we believe Haeberli (1985) does not require any introductory context.

Provide a concise clarification or rationale for merging active and inactive landforms into the "intact" category, referring to Haeberli (1985) and Barsch (1996) as appropriate.

We are plainly adopting the traditional classification scheme proposed by Haeberli (1985) and Barsch (1996), which forms an international benchmark. Consequently, the merging follows the rationale

according to which, intact (active + inactive) rock glaciers would contain permafrost, as described in the original manuscript.

Also clarify the rationale behind choosing the 1 cm displacement threshold and its relevance to the study's time frame.

The rationale is taken from Barsch (1996) and the relevant thresholds for dynamic classification (lines 130-138). In the context of our work, the 1 cm threshold fits the sensitivity of Sentinel-1 InSAR, which has been assessed to be about 6–7 mm in a major validation project over urban areas (Crosetto, M.; Monserrat, O.; Bremmer, C.; Hanssen, R.; Capes, R.; Marsh, S. Ground motion monitoring using SAR interferometry: Quality assessment. Eur. Geol. 2009, 26, 12–15). A comparable high degree of coherence over a multiannual period is typically recorded over rock glaciers, as shown by Strozzi et al. (2020). In the revised version of the manuscript, explicit reference to Crosetto et al (2009) and Strozzi et al (2020) has been made in the closing sentence of section 3.1.

3.2

The mention of the InSAR-based kinematic characterization is clear but consider providing a summary or overview of the methodology in this section.

We thank the referee for this comment. As mentioned at the outset of this rebuttal letter, the methodology adopted in our study builds on detailed descriptions provided in Bertone et al (2022) and in RGIK (2023). Given the composite structure of the current manuscript, we think that duplicating such a detailed, technical description of the methods would make the paper bulkier and more difficult to read for the broader readership of the journal.

Ensure consistent terminology throughout the document. For example, if "kinematic characterization" and "kinematic classification" are used interchangeably, clarify their relationship, or use the term consistently.

Characterization holds a more general connotation that distinguishes rock glaciers into moving, not moving and undefined. Classification refers to the assignment of a specific kinematic class to a moving area or to a rock glacier.

3.2.1

Specify the exact periods in September to October of 2018 and 2019 when the acquisitions were made. Providing specific dates or a more precise timeframe would enhance clarity.

Specific dates bounding the study period are reported in line 226 of the original manuscript (i.e., between 02 Sept 2018 and 23 Oct 2019). To address the reviewer's concern, we have copied these dates in lines 165-166 of section 3.2.1.

While the document is likely targeted at a technical audience, it's important to define technical terms or acronyms, such as LOS (Line of Sight)

Please note that Line Of Sight has been defined in line 185.

When discussing underestimation of ground deformation due to the LOS, provide a brief explanation of the consequences or implications of such underestimation in the context of the study.

We would like to thank the reviewer for this comment. Discussion on signal underestimation and cascading effects on the rock glacier kinematic characterization is developed for some length in sections 5.1 and 5.2 of the Discussion (e.g., Figure 11a; Figure 12 and relevant caption; Figure 13 and relevant caption). To address the reviewer's concern and improve the logical linkage between methods, results, and discussion, we have added the following sentence at line 198 in 3.2.1: "Evaluation of signal underestimation for all the

rock glaciers inventoried in this study, as well as site specific examples and relevant practical implications will be discussed in sections 5.1 and 5.2".

If appropriate, consider using subheadings in the section discussing limitations and challenges of InSAR measurements.

The three major imitations are listed and described in Lines 176-187. In our view, splitting these eleven lines of text into three subheadings does not seem appropriate and would break the logical flow of the relevant paragraph.

Please use consistent units for displacement measurements (e.g., cm, meters) to avoid confusion.

The units of annual displacement rates adopted in this paper for the kinematic classification of moving areas and rock glaciers follow the specifics detailed by Bertone et al (2022) and by RGIK (2023). The use of cm, dm, and m aims to convey contrasting orders of magnitude of displacements to scientists, as well as to practitioners dealing with risk assessment around infrastructure.

Following the referee's suggestion, we have revised the entire manuscript and rectified relevant unit notations that appeared inconsistent.

In Equation 1, ensure that the variable $U$ is defined or referred to in the text, so readers can understand its significance.

Thank you for this comment. U stands for underestimation. In line 188, we have added U in brackets after "underestimation".

3.2.2

Provide a brief explanation of why the minimum area threshold of 20 adjacent pixels is chosen. What significance does this threshold have, and how does it contribute to the consistency and accuracy of the inventorying process?

A threshold of at least 20 to 30 pixels is used to ensure clear identification and representativeness of the signal and to avoid sporadic noise that can occur on portions of terrains not related to displacement. This requirement complies with the guidelines established in Bertone et al (2022), and RGIK (2023). To address the reviewer's comment, we have rewritten the sentence as follows, explicitly stating the source of the criterion adopted (lines 237-239):

"To ensure that different operators outline moving areas consistently, as well as to distinguish moving areas from surrounding noise in a given interferogram, a minimum area threshold is applied. Following Bertone et al. (2022) and RGIK (2023) guidelines, moving areas need to involve at least 20 adjacent pixels in a gridded interferogram."

Elaborate on the criteria used for assigning velocity classes based on the change in colour observed on the wrapped interferograms. Explain how these classes are indicative of different movement rates, and clarify any assumptions made during this classification.

We thank the referee for this comment. As mentioned at the outset of this rebuttal letter, the methodology adopted in our study builds on detailed descriptions provided in Bertone et al (2022) and in RGIK (2023). We believe this request goes beyond the scope of the paper, since we are adopting a methodology that has been around for some time. For more information and justification of the kinematic classes adopted, please refer to Barboux et al (2014), Bertone et al (2022) and RGIK (2023).

Briefly discuss the limitations associated with the choice of a maximum temporal baseline of one year and its implications for velocity detection. This would help readers understand the study's constraints and potential sources of uncertainty.

This part has been described in section 3.1. In the context of our work, the 1 cm threshold fits the sensitivity of Sentinel-1 InSAR annual interferograms, which has been assessed to be about 6–7 mm in a major validation project over urban areas (Crosetto, M.; Monserrat, O.; Bremmer, C.; Hanssen, R.; Capes, R.; Marsh, S. Ground motion monitoring using SAR interferometry: Quality assessment. Eur. Geol. 2009, 26, 12–15). A comparable high degree of coherence over a multiannual period is typically recorded over annual interferograms on rock glaciers, as shown by Strozzi et al. (2020). In the revised version of the manuscript, explicit reference to Crosetto et al (2009) and Strozzi et al (2020) has been made in the closing sentence of section 3.1. Uncertainty associated with this time window are discussed at the entire inventory level and at the site-specific level respectively in section 5.1 and 5.2

Ensure consistency in units throughout the section, especially when expressing velocity classes. For example, use consistent units such as cm/yr for clarity.

Thank you for spotting this inconsistency. Accordingly, we have modified units in Figure 5 and have adopted cm yr-1 dm yr -1 and m yr-1 (as opposed to using interchangeably /yr and yr-1) to make them consistent with the other tables and figures.

3.2.3

Provide explicit criteria for classifying rock glaciers into the categories of moving, not moving, and undefined.

We believed the criteria outlined in lines 259-262 were explicit and straight forward.

To address the referee's concern, we have rephrased the relevant set of sentences as follows (line 259 and following): "In particular, based on the presence/absence of at least one moving area (i.e., $\approx$ 1 cm yr-1) on annual interferograms, within the geomorphologic footprint of a given rock glacier, we classify rock glaciers into *moving*, *not moving*, and kinematically *undefined*. Accordingly, a rock glacier polygon that encloses one or more moving areas is labelled as moving. A rock glacier that does not exhibit any detectable moving area is labelled as not moving. A rock glacier for which reliable kinematic information is not available -- due to extended layover, shadowing, atmospheric artefacts, phase bias, or decorrelation -- is labelled as undefined."

Elaborate on the rationale behind assigning the median class to a rock glacier hosting multiple moving areas with diverse kinematics. Explain why more weight is given to the largest moving area closest to the rock glacier front.

As mentioned at the outset of this rebuttal letter, the methodology adopted in our study builds on detailed descriptions provided in Bertone et al (2022) and in RGIK (2023). Given the composite structure of the current manuscript, we think that duplicating such a detailed, technical description of the methods would result in a bulkier paper, whose logical flow would become much more difficult to follow by the broader readership of the journal.

To address the referee's concern, in the revised version of the manuscript, we have made explicit reference to Bertone et al. (2022) and to RGIK (2023) and have provided explicit justification for giving more weight to the moving area located in proximity of the rock glacier front. The revised text reads as follows (line 267 and following): "Wishing to minimize inter-operator variability, to a rock glacier that hosts two equally dominant moving areas with diverse kinematics, we assign the category of the area closer to the front. In

case of multiple, equally dominant moving areas spanning across a wide range of categories, the rock glacier is assigned median kinematic category (RGIK, 2023)."

Justify the chosen categories for kinematic classification (e.g., cm/yr, cm to dm/yr, etc.). Provide a rationale for these divisions and discuss how they contribute to the understanding of rock glacier dynamics.

As per prior comment, the kinematic categories follow the specifics detailed by Bertone et al (2022) and by RGIK (2023). Subdivision into discrete categories is meant to account for the intrinsic uncertainty associated with the assignment of a kinematic attribute to a rock glacier surface, where the rock glacier surface rarely matches the extent of a single moving area.

Discuss any temporal considerations in the kinematic classification process. For instance, mention if the classification is based on an annual average or if there are seasonal variations considered.

The referee is right. Our text was not clear enough. In section 3.1 we stated that our interferograms refer to snow free periods, but did not follow up specifying that consequently, our annual rates of displacements refer to snow free periods as well. To address the referee's concern, we have enriched the methodological description in lines 265-266. The revised sentence reads as follows: "Categorical assignment considers the seasonal variability of rock glacier downslope deformation, generally highest during late summer and autumn months (Delaloye and Staub, 2016; Kellerer-Pirklbauer et al., 2018). Our annual rates of surface deformation reflect kinematic information retrieved from redundant interferograms computed over snow free periods (i.e., from late July to late October). This kinematic approach, which averages across a minimum of two-to-three months, ensures that velocity peaks are not inadvertently selected as representative annual rates of surface displacement."

Clearly state the sources of data or observations used for the kinematic analysis. If there are specific interferograms or datasets used in this analysis, refer to them explicitly.

The interferograms used in this study are reported in Figure 1b. In addition, the list of interferograms utilized in this study is reported in the newly developed **Supplementary Table S1** (see **page 17** in this document).

Address potential sources of uncertainty in the kinematic classification process. Discuss how factors like layover, shadowing, atmospheric artifacts, phase bias, or decorrelation.

As indicated in the manuscript, if reliable kinematic information is not available due to extended layover, shadowing, atmospheric artefacts, phase bias, or decorrelation, we classify as undefined. At line 305 we also quantify this class (18%).

Results

The text is generally well-structured and logically organized. Consider providing a brief introductory paragraph to set the context and purpose of the study before delving into specific sections.

Thank you for your comment. The structure of the results is introduced in a brief introductory paragraph (lines 274-278). The purpose of the study (i.e., three specific objectives) has been stated explicitly in the introduction and doesn't seem appropriate to repeat it here. To establish a direct connection with the objectives, we have added an opening sentence to section 4, which reads as follows: "To pursue the objectives detailed in section 1, we have subdivided the results into three subsections. Accordingly, we begin …".

Given the earlier suggestion of adding a guided structure of the entire paper at the end of the introduction (section 1) – a suggestion that we have accepted in one of our earlier replies -- we consider redundant adding further introductory statements.

There is a slight repetition in phrases like "moving areas" and "rock glaciers." Consider using synonyms occasionally for better readability.

We believe this is a matter of style. We prefer to stick to the actual terms and avoid synonyms that may confuse the reader. We hope our choice will be acceptable.

Ensure consistent use of terminology. For instance, the text uses both "rock glacier fronts" and "rock glacier polygons." Clarify if these terms refer to distinct features or are used interchangeably.

The front forms the frontal wall of a rock glacier (also termed terminus by some authors). The rock glacier polygon ecloses the entire landform. The rock glacier front is included within the polygon. We believe this is common knowledge for the broader rock glacier and periglacial geomorphology communities.

Given the complexity of InSAR analysis, consider adding a brief explanation or reference for readers who may not be familiar with the technique. This can enhance the accessibility of your findings.

We are not completely sure that we have fully understood the meaning of this comment. Please advise and let us know if our approach is acceptable.

Key references that introduce InSAR analysis are reported in two paragraphs of the introduction (section 1):

1) Lines 70-79: "when wishing to upscale the assessment of 70 rock glacier activity over entire basins, or regions, satellite Synthetic Aperture Radar Interferometry (InSAR) may prove fundamental. As a consolidated technique for detecting and mapping land surface deformation at suitable spatial and temporal resolution (Rosen et al., 2000), InSAR warrants an objective kinematic characterization (hence dynamic classification) of rock glaciers over large areas (e.g., Liu et al., 2013; Necsoiu et al., 2016; Wang et al., 2017; Bertone et al., 2019; Brencher et al., 2021; 75 Reinosch et al., 2021). In this context, we argue that the kinematic approach – developed by Barboux et al. (2014) and refined by Bertone et al. (2022) – entailing the detection and delineation of moving areas (i.e., areas of detectable surface deformation on wrapped interferograms), besides elucidating which rock glaciers move, and consequently bear permafrost, may prove strategic for documenting where about the rock glacier (e.g., the main front, the rooting zone, or the entire landform) and in what proportion surface deformation occurs."

2) Lines 81-85: " Although the application of InSAR technology to rock glacier inventories holds straight forward advantages, amply demonstrated for single, a cluster, or many rock glaciers (e.g., Barboux et al., 2014; Strozzi et al., 2020; Lambiel et al., 2023; Bertone et al., 2023), a systematic and quantitative evaluation of the improvements afforded to a traditional "geomorphologic" inventory, encompassing both intact and relict landforms over broad spatial scales, is missing."

Also, emphasize the significance of integrating the geomorphologic and InSAR approaches earlier in the section to highlight the study's methodology and its contribution to understanding rock glacier dynamics.

Integration of geomorphologic and INSAR approaches is stated explicitly in the title of the manuscript and forms the main objective of the paper:

Lines 86-92: "To address this gap, following operational guidelines on the InSAR-based kinematic characterization of rock glaciers jointly proposed by ESA Permafrost CCI and IPA Action Group on rock glacier inventories (Bertone et al., 2022), we wish to integrate geomorphologic and InSAR-based inventorying approaches in selected valleys of western South Tyrol, where rock glacier occurrence is overwhelming. Therein, starting from the compilation of a geomorphologic inventory, we aim to: (i) characterize the kinematics of InSAR-based moving areas and relevant hosting rock glaciers; and (ii) evaluate InSAR-derived improvements, in terms of inventory completeness and uncertainty reduction, in the detection and dynamic classification of rock glaciers."

The significance of integrating geomorphologic and InSAR-based approaches is discussed in the discussion of the results (section 5.1):

Lines 421-429: "In this contribution, we illustrate the importance of complementing a geomorphologic rock glacier inventory with InSAR-derived kinematic information. Integration of the two inventorying approaches, each of which characterized by specific strengths and weaknesses, allows reducing the overall uncertainty associated with the procedures of rock glacier detection, mapping and dynamic classification. On one hand, visual interpretation of multi-temporal optical imagery is crucial for outlining the morphological footprint of a rock glacier, and serves as a benchmark for developing automated mapping routines (e.g., Robson et al., 2020; Reinosch et al., 2021); on the other hand, visual interpretation of multi-temporal wrapped interferograms – except where decorrelation occurs – ultimately determines whether a rock glacier exhibits surface deformation, at what mean annual rate, and in which portion of its morphological footprint, thus confirming or rectifying prior evaluation solely based on the interpretation of morphological attributes."

Integration of geomorphologic and InSAR-based approaches leads to the dynamic reclassification of "geomorphologic" rock glaciers, as shown in Table 4, Figure 5c and Figure 11b and discussed in the second half of section 5.1. Finally, the significance and the practical implications of this methodological integration are shown from site-specific case studies (Figures 12 and 13) up to the entire study area (Figure 14) in section 5.2. We think that section 4 (the Results) should plainly describe observed patterns, while avoiding digressing into interpretations and implications. Please let us know if our approach is acceptable.

Maintain consistency in the use of abbreviations and acronyms throughout the text. Ensure that abbreviations are defined upon first use.

A check for consistency has been performed throughout the entire document.

The analysis of the altitudinal distribution of moving areas is clear. However, provide more context or potential explanations for the observed dependence on slope aspect and the remarkable gap on south-facing slopes for faster moving areas.

We believe that this section (section 4) should host the plain description of the results, avoiding to mix results with interpretations, or possible explanations. The latter are dealt with in the discussion (section 5). To address the reviewer's comment, we briefly report a general explanation, supported by a number of previous studies, as follows (lines 343-345): "The altitudinal distribution of moving areas displays a progressive increase from north- to south-facing slopes (Fig. 7), grossly confirming the strong dependence on slope aspect previously suggested by the qualitative geomorphologic approach (Fig. 4a). This pattern, previously observed across the Alps and elsewhere (e.g., Barsch, 1996; Imhof, 1996; Scapozza and Mari, 2010; Scotti et al., 2013; Falaschi et al., 2015; Wagner et al., 2020), is regarded as the expression of aspect-dependent surface and subsurface thermal conditions, that is, rock glaciers occur at comparably higher elevations at slope aspects where potential incoming solar radiation is higher."

Scapozza, C., Mari, S., 2010. Catasto, caratteristiche e dinamica dei rock glaciers delle Alpi Ticinesi. Bollettino della Società Ticinese di Scienze Naturali 98, 15–29.

The analysis of the size of moving areas is well-presented. Consider briefly discussing the implications of the distinctive peak in median size for areas moving at 30-100 cm yr-1.

Thank you for this comment. We do not see major implications for this finding. To address the reviewer's comment, we have added the following sentence in lines 359-360: "This finding suggests that where such moving areas occur, they are likely to dictate (or influence heavily) the kinematic class assigned to the hosting rock glacier."

Discuss the absence of an obvious correlation between moving area size and slope aspect. Elaborate on the implications of this finding and how it aligns or contrasts with expectations.

Being the first quantitative study in the moving areas in the literature, we had no expectations. Similarly, we do not foresee any implication linked to this finding. We would be glad to add specific implications in section 5, should the referee have something to add on this matter.

The observation that moving area size does not appear to correlate with elevation is notable. Provide some insight or hypothesis on why this might be the case and discuss its significance.

Being the first quantitative study in the moving areas in the literature, we had no hypothesis beforehand. Similarly, we do not foresee any implication linked to this finding. We would be glad to elaborate on the significance in section 5, should the referee have some specific suggestions on this matter.

Discuss the factors that might contribute to the observed variability in total moving area below 2500 m a.s.l. Provide possible explanations for cases where the entire rock glacier surface moves versus cases with less than 10% moving extent.

We believe that this section (section 4) should host the plain description of the results, avoiding to mix results with interpretations, or possible explanations. Following this logic, the variability of total moving area as a function of elevation is discussed at the end of section 5.2 and is further elaborated in section 6 (the conclusions). Our interpretation relates the altitudinal dependence of total moving area to the spatial distribution of permafrost, which is sporadic at low elevations and increases progressively with increasing elevation, becoming discontinuous.

Accordingly, in section 5.2 (lines 569-586) we discuss the following:

"Equally important, from a basic process-oriented standpoint, InSAR information proves fundamental for imaging how this altitudinal transition manifests through changing rates and styles of rock glacier surface deformation. Collectively, the systematic variability of total moving area suggests that as rock glaciers move faster, an increasingly larger proportion of their surface gets kinematically involved, and that this proportion increases with elevation (Figures 9 and 10). This interpretation becomes apparent when umping MA variability into rock glacier units, that is, when representing rock glacier annual velocity and percent MA cover (i.e., the proportion of a rock glacier surface occupied by moving areas) as a function of elevation (Fig. 15). Indeed, both variables increase linearly with elevation up to the 2600-2800 m 575 band, beyond which an inflection occurs and consistent average values (and ranges of variation) are attained. According to the classification scheme recently proposed by RGIK (2022), the average dynamic state that characterize rock glaciers below and above 2600 m are consistent respectively with transitional and active rock glacier types. Our findings differ from similar work conducted in the Uinta Mountains (Utah), in which no apparent altitudinal dependence of rock glacier velocity was found (Brencher et al., 2021). Most importantly, the altitudinal inflection falls between the -1°C (2595 m) and -2 °C MAAT (2765 m) – regarded as the lower boundary for discontinuous permafrost occurrence (Haeberli, 1983; Haeberli et al., 1989) – and appears to be decoupled from local topographic conditions, since average rock glacier slope gradient stays about constant (i.e., around 0.45 m/m) across the entire elevation range (Fig. 15). Based on these observations, we argue that the altitudinal increase in average rock glacier velocity and average MA cover represents the dynamic expression of increasing permafrost distribution (i.e., a transition from sporadic to discontinuous), until optimal thermal conditions are reached above 2600 m.

This interpretation is further developed in the second part of the conclusions (section 6, lines 607-620):

"The existence of a broad altitudinal transition zone, in which intact and relict rock glaciers coexist, reinforces the need of using InSAR technology for the site-specific characterization of rock glacier displacement across large spatial scales, hence for inferring the spatial distribution of discontinuous-tosporadic permafrost in mountain terrain. In this context, integration of the two inventorying approaches gains additional importance from a basic, process-oriented standpoint, as it proves particularly effective for imaging how creeping of perennially frozen, ice-debris mixtures varies (in intensity and spatial distribution) as a function of elevation. Specifically, we find that average annual rates of rock glacier surficial displacement and percent MA (moving area) cover – where the latter variable relies both on the morphological delineation of rock glacier footprints in optical imagery and the kinematic characterization of hosted moving areas in wrapped interferograms – increase linearly with elevation until a sharp inflection occurs and consistent average values are attained above 2600 m a.s.l. Considering that this altitudinal inflection occurs right between the -1°C and -2 °C MAAT – regarded as the lower boundary for discontinuous permafrost occurrence – we interpret this altitudinal pattern as the geomorphic signature of increasing permafrost distribution (i.e., transition from sporadic to discontinuous), until optimal thermal conditions, which promote "full scale" viscous creep and shearing at depth, are met."

Offer insights into the observed scatter for rock glaciers moving at cm annual rates and the abrupt drop in scatter for faster rock glaciers. Relate these observations to the overall dynamics and behaviour of rock glaciers.

Similarly to what stated in our previous reply, our interpretation relates the altitudinal dependence of moving area velocity and spatial distribution to the spatial distribution of permafrost, which is sporadic at low elevations and increases progressively with increasing elevation, becoming discontinuous. However, we cannot discuss this matter in section 4, as this is topic for the discussion. Accordingly, this part is elaborated in section 5.2 and in the conclusions.

To address the referee's concern, we have added the following sentence (lines 393-395): "As will become clear, the increasing altitudinal clustering of moving areas characterized by higher velocity, imparts a characteristic altitudinal continuum in rock glacier dynamics, which we consider geomorphic expression of sporadic-to-discontinuous permafrost transition (section 5.2)."

Consider concluding the section of the results with a brief transition or summary statement that leads into the future part of the study. This will help maintain a smooth flow in the narrative.

We would like to thank the referee for this comment. We believe this is a matter of writing style. In this respect, please note that lines 401-419 (section 5) form a transition between the plain description of the results (outlined in section 4) and the detailed discussion of specific aspects (developed in sections 5.1 and 5.2). Consistently with what stated in some of the foregoing replies, we think that Results and Discussion should stay, for as much as possible, carefully separated from each other. We hope the referee could share our view on this.

Discussion

Compilation of a rock glacier inventory is crucial for understanding challenges associated with the changing high-mountain cryosphere. It will be better if the authors can add aids in evaluating responses to warmer climate conditions, from transitioning to relict morphologies to destabilization with increased velocity.

Aids in evaluating responses to warmer climate conditions were included in the second sentence of section 5 (lines 402-408), to make the importance of using rock glacier velocity (RGV) time-series more explicit, we have rewritten the sentence in the following way:

"For example, detecting topographic change and analysis of rock glacier velocity (RGV) time-series allows evaluating contingent responses to warmer climate conditions, which may range from transition to relict morphologies, with widespread surface subsidence and loss of downslope momentum (Bollmann et al., 2015; Necsoiu et al., 2016), to generalized rock glacier destabilization with orders of magnitude increase in mean annual velocity at their fronts (Delaloye et al., 2008; Scotti et al., 2017a; Marcer et al., 2019; 2021;

Pellet et al., 2023; RGIK, 2023b; 2003c) and enhanced potential for debris-flow occurrence (Kummert and Delaloye, 2018; Kummert et al., 2018; Kofler et al., 2021)."

Please discuss potential biases introduced by the integration of InSAR data, such as the reliance on elevation for dynamic classification. Are there alternative methods to mitigate such biases.

We think there was a misunderstanding. The bias we are referring in the manuscript (lines 479-482, section 5.1) relates to the geomorphologically-based dynamic classification of rock glaciers (i.e., classification based on visual interpretation of optical images). In this respect, we show that our InSAR-based analysis aids reducing this bias, though an objective (as opposed to a subjective one) dynamic reclassification of rock glaciers into moving and not moving features.

Please see lines 479-482: "InSAR contribution allowed lessening what appears to be a systematic, altitudinal-driven bias that may affect operators' visual interpretation of landforms. Our findings suggest that operators, to some extent, may base their dynamic classification of rock glaciers – especially when these do not exhibit unambiguous morphological evidence – on the elevation at which they are located."

To recap, our response to the comment is that InSAR-based analysis can mitigate such altitudinal-driven bias that appears to influence operators' judgment.

Ensure that the citations are up-to-date, and if possible, include the most recent studies in the field to strengthen the discussion.

Citations have been updated.

Guidelines prepared by the IPA Working Group on Rock Glacier Inventories and Kinematics (RGIK) have been updated to the latest version:

RGIK, 2023a. Guidelines for inventorying rock glaciers: baseline and practical concepts (version 1.0). IPA Action Group Rock glacier inventories and kinematics, 25 pp, doi:10.51363/unifr.srr.2023.002

Reference to the full conference paper on the geomorphologic rock glacier inventory in South Tyrol has been added:

Scotti R, Mair V, Costantini D, Brardinoni F. In press. A high-resolution rock glacier inventory of South Tyrol. Evaluating lithologic, topographic, and climatic effects. Full paper, 8 pp. ICOP 2024, 16-20 June 2024, Whitehorse, Yukon, Canada.

Reference to rock glacier velocity (RGV) time series:

Pellet, C., Bodin, X., Cusicanqui, D., Delaloye, R., Kääb, A., Kaufmann, V. Noetzli, J., Thibert, E., Vivero, S. and Kellerer-Pirklbauer, A. 2023. Rock Glacier Velocity. In Bull. Amer. Meteor. Soc. Vol104(9), State of Climate 2022, pp 41-42, doi:10.1175/2023BAMSStateoftheClimate.1

RGIK, 2023b. Rock Glacier Velocity as an associated parameter of ECV Permafrost: Baseline concepts (Version 3.2). IPA Action Group Rock glacier inventories and kinematics, 12 pp.

RGIK, 2023c. Rock Glacier Velocity as associated product of ECV Permafrost: practical concepts (version 1.2). IPA Action Group Rock glacier inventories and kinematics, 17 pp.

Kellerer-Pirkbauer, A, et al., 2024. Acceleration and interannual variability of creep rates in mountain permafrost landforms (rock glacier velocities) in the European Alps in 1995–2022. Environ. Res. Lett., 19, 034022. https://doi.org/10.1088/1748-9326/ad25a4.

Conclusion

Authors, please acknowledge the contribution of InSAR to geomorphologic inventory reliability.

This acknowledgement is stated in the opening sentence of the Conclusions (lines 591-593): "In this paper, we show that the integration of InSAR-based information can improve the reliability of a geomorphologic inventory both in terms of completeness, through detection of additional intact rock glaciers on wrapped interferograms, and in terms of dynamic classification, through uncertainty reduction. "

Authors, please address potential limitations or challenges associated with future work.

Thank you for this comment. We have revised and enriched text in lines 598-602. The modified text is the following: "In this context, the next challenge that awaits similar studies in south-western South Tyrol and elsewhere, will be that of reducing the number of undefined rock glaciers by considering longer baselines (e.g., 2 to 3 years) in Sentinel-1, and by integrating constellations characterized by different LOS geometry and SAR band (e.g., CosmoSkyMed, SAOCOM and TerraSAR-X). We expect that this combination of sensors would warrant greater flexibility at covering all slope aspects, while lowering the 1 cm yr-1 threshold of (assuredly) detectable deformation.

Provide further clarification on the implications of the increase in altitudinal overlap between relict and intact rock glaciers.

The implications for the increase in altitudinal overlap are reported in lines 603-610. To address the referee's concern, we have made the main implication more explicit (i.e., drawing an altitudinal limit for the occurrence of discontinuous permafrost becomes a more difficult and less straight forward task). The revised sentence now makes the practical implication of our finding explicit (line 607): "As practical implication, the existence of a broad altitudinal transition zone, in which intact and relict rock glaciers coexist, reinforces the need of using InSAR technology for the site-specific characterization of rock glacier displacement across large spatial scales, hence for inferring the spatial distribution of discontinuous-to-sporadic permafrost in mountain terrain."

The conclusions section provides valuable insights into the contributions of InSAR technology and the implications for understanding rock glacier dynamics. Addressing the suggestions can further enhance the clarity and impact of the conclusions.

Comment on Figures

Figure 1: The horizontal line graph showing the temporal baselines for both ascending and descending passes is cluttered. I suggest you to replace the line graph with a bar graph with dates along the x axis and the baseline days along the y axis. Easier to interpret that way. Also panel a map doesn't show any useful information. Can you instead zoom into the two valleys instead and show a DEM or even a Sentinel-1 backscatter map. That would set the InSAR context.

We thank you the reviewer for these comments. To address the reviewer's concern on InSAR information completeness/readability/clarity (Figure 1b), we have added **Supplementary Table S1** (see next page) detailing the characteristics of all interferograms utilized.

Figure 1b wishes to convey **continuous coverage** and **redundance** of snow free periods from 02 Sept 2018 to 23 Oct 2019 (these two dates were also added in the revised version of the figure caption). Using bar graph, as opposed to the current line graph, we believe it is just a question of style that does not influence readability. We hope that by adding Suppl Table S1 we have solved this issue.

Regarding Figure 1a, this figure is meant to illustrate an orogen scale view of the European Alps, providing context to the study area and its inner position within the Alps. The footprints of the two tracks utilized in this work are equally important, as they show that the study area falls well within each track, and that no edge effects involved in the acquisitions. In the original manuscript, wishing to avoid duplicating

cartographic information on the study area, we did not add a close-up view, considering that three maps with elevation bands and shaded relief background are reported in Figure 5 (panels a, b, and c).

**Supplementary Table S1.** Information on the Sentinel-1 interferograms used in this study.

| Master image (date) | Slave image (date) | Temporal baseline (days) | Orbit |
|---|---|---|---|
| 04/09/2018 | 16/09/2018 | 12 | Ascending |
| 16/09/2018 | 22/10/2018 | 36 | Ascending |
| 28/09/2018 | 04/10/2018 | 6 | Ascending |
| 28/09/2018 | 10/10/2018 | 12 | Ascending |
| 28/09/2018 | 22/10/2018 | 24 | Ascending |
| 04/10/2018 | 10/10/2018 | 6 | Ascending |
| 04/10/2018 | 11/09/2019 | 342 | Ascending |
| 10/10/2018 | 05/09/2019 | 330 | Ascending |
| 11/09/2019 | 23/10/2019 | 42 | Ascending |
| 17/09/2019 | 17/10/2019 | 30 | Ascending |
| 29/09/2019 | 11/10/2019 | 12 | Ascending |
| 29/09/2019 | 23/10/2019 | 24 | Ascending |
| 11/10/2019 | 17/10/2019 | 6 | Ascending |
| 11/10/2019 | 23/10/2019 | 12 | Ascending |
| 02/09/2018 | 08/09/2018 | 6 | Descending |
| 02/09/2018 | 26/09/2018 | 24 | Descending |
| 02/09/2018 | 08/10/2018 | 36 | Descending |
| 08/09/2018 | 14/09/2018 | 6 | Descending |
| 08/09/2018 | 20/09/2018 | 12 | Descending |
| 14/09/2018 | 08/10/2018 | 24 | Descending |
| 14/09/2018 | 20/10/2018 | 36 | Descending |
| 26/09/2018 | 08/10/2018 | 12 | Descending |
| 08/10/2018 | 14/10/2018 | 6 | Descending |
| 08/10/2018 | 20/10/2018 | 12 | Descending |
| 14/10/2018 | 21/09/2019 | 342 | Descending |
| 26/10/2018 | 03/10/2019 | 342 | Descending |
| 03/09/2019 | 15/09/2019 | 12 | Descending |
| 03/09/2019 | 09/10/2019 | 36 | Descending |
| 15/09/2019 | 21/09/2019 | 6 | Descending |
| 21/09/2019 | 15/10/2019 | 24 | Descending |
| 03/10/2019 | 15/10/2019 | 12 | Descending |
| 09/10/2019 | 15/10/2019 | 6 | Descending |

Figure 2: Can the schematic be better shown? The orange boxes show a show of sub parameters and variables (which is OK and is explained in text) that could be better represented In the flowchart. Especially the classification part from 'moving areas' to 'rock glaciers' box where you show multiple lines characterizing different moving areas at different velocities in cm/yr. to rock glacier velocities in different units other than only cm/yr. (see one of my comments above regarding this).

The classification from moving areas to rock glaciers plainly adheres to the specifics detailed by a panel of international experts, which forms the methodological foundations of our work (Bertone et al., 2022; RGIK, 2023).

Such methodology has been approved by the extended IPA Rock Glacier Working Group on Rock Glaciers and Kinematics (RGIK), which includes more than one hundred international experts. The scope of this initiative was to propose and summarize a set of shared guidelines to ensure the compilation of **consistent** rock glacier inventories worldwide. To avoid creating confusion in the community, we believe we must adhere to the conceptual, categorical and graphical rules (for example the linkage from kinematic moving area classes to kinematic rock glacier classes) established in the RGIK guidelines.

Figure 3: I see a few issues here. Showing the whole area 'including' RG1 and RG2 masks a lot of internal information within RG1 and RG2. To show fringes more clearly I would strongly suggest to redo the figure showing only the RG1 and RG2 ROIs as it is and remove the other areas. Secondly, how do you delineate regions in terms of no fringes, noisy one, decorrelated ones and complete fringes? I suggest to remove those tags from the figures and let the readers understand from the text. The legend used for the fringe scale needs to be changed. The reader now has to use their 'math' brain to calculate the displacement. Can we make it simpler and use discrete range of values from RG1 and RG2 ranges?

To address the referee's comment, we have added **Supplementary Figure S1** (see next page), which contains a clean version of Figure 3, with no polygons (except RG1 and RG2), and legend.

Please consider that Figure 3 is structured as a working example to walk the reader through the different phases of moving area (MA) delineation, moving area kinematic classification, and subsequent rock glacier kinematic classification. Removing the MA outlines will leave the reader lost or having to guess what to do next. The tags are especially critical to guide the reader across examples of fringe, decorrelated and plain (no fringe) patterns. We wish to emphasize that moving areas (and the moving area outlines) are associated with phase changes due to displacements in the interferograms (i.e., fringe patterns). Consequently, plain and decorrelated patterns are not enclosed by polygon outlines. Furthermore, the figure shows only a few interferograms, but delineation and classification of MA is based on a larger number of interferograms that is not possible to include within a single figure (see Figure 1b and **Supplementary Table S1**)

We would like to clarify that in this figure we are plainly illustrating and adopting the methodology (both in terms of classification and graphical rules) established and detailed in Bertone et al (2022), and in RGIK (2023). Such methodology has been approved by the extended IPA Rock Glacier Working Group on Rock Glaciers and Kinematics (RGIK), which includes more than one hundred international experts. The scope of this initiative was to propose and summarize a set of shared guidelines to ensure the compilation of **consistent** rock glacier inventories worldwide. On this premise, to avoid creating confusion in the international community, we believe we must adhere to the conceptual, categorical and graphical rules (for example the color coding in the interferograms) established in the RGIK guidelines.

[Figure]

**Supplementary Figure S1** (Clean version of Figure 3). Sample kinematic characterization applied to two rock glaciers in Ultimo Valley 200 showing (a) initial manual delineation of rock glacier polygons (black linework) on optical imagery; Sentinel-1 interferograms calculated over (b) 6 days (2018/09/28 – 2018/10/04), (c) 12 days (2019/10/11 – 2019/10/23), (d) 24 days (2018/09/28 – 2018/10/22), and (e) 330 days (2018/10/10 – 2019/09/05). In panel f is reported the color-coded scheme used for evaluating phase difference on interferograms. Orthoimage from the Autonomous Province of Bolzano (https://geoportale.retecivica.bz.it/geodati.asp; last access: June 2023).

Figure 4a and 7: I am slightly confused by the directions here in the x-axis. Where is the origin here?

Strictly speaking, the north (N) represents some sort of origin (0 degrees). The x-axis represents slope aspect and therefore varies from 0 to 360 degrees.

What I mean is when you say N, where is North with respect to RG1 and RG2. The y axis label is RG front elevation. So RG means both Rock Glaciers or just one? Please clarify.

RG1 and RG2 are not part of this figure. Figure 4a represents the altitudinal distribution of all rock glacier fronts (as mapped in the geomorphological inventory) plotted as a function of slope aspect. RG is a

contraction for rock glacier (as much as MA is a contraction for moving area). The figure represents the altitudinal distribution of all mapped rock glaciers. In each aspect category, the filled symbol (a circle for the intact and a diamond for the relict) indicates the median elevation of all the rock glaciers in that given aspect class. Bars enclose interquartile range (25% to 75% of the altitudinal frequency distribution). Please see figure caption.

To improve readability, in the caption of Figure 4a we have added "(filled circles and diamonds)" after the word "median".

And what I miss in Figure 7 is also this spatial reference with respect to which glacier you are referring to. Maybe a small inset map showing the direction could help?

The scatter plot in Figure 7 depicts all the moving areas mapped in this study. Each dot corresponds to a moving area. There is no particular spatial reference involved.

Is Figure 6 MISSING?

Figure 6 is located in page 14.

Figure 11: It seems like in panel b, there are values above 3000m masl, but the secondary Y-axis only shows range up to 3000m. In panel a, what is slope (m/m) what is m/m?

Thank you for this comment. The pace of the y-axis is constant. Therefore, the top and lowermost altitudinal labels are 3200 and 1600 m a.s.l. To address the reviewer's concern, we have added tick marks and labels related to 1600 m and 3200 m a.s.l.

Slope = rise/run, both expressed in meters (i.e., 0.1 m/m equals a 10% slope gradient). It is the classical way to express slope in geomorphology.

Figure 14: The same problem as above with the y axis masl ranges in both lower and higher ends. Also the box plots with the sample numbers tagged above looks cluttered and busy. What is the relevance of sample numbers in this figure? If its not used in the analysis, I suggest you to remove them.

As per prior comment, the pace of the y-axis is constant. Therefore, the top and lowermost altitudinal labels are 3100 and 1700 m a.s.l. Relevant missing labels have been added.

When presenting boxplots, it is critical to show the sample size they are based on. The scope of including the number of observations above each box-whisker is that to inform the reader on the relevant statistical robustness. Following this logic, a greater number of observations implies a more robust statistical representation. Indeed, this type of information is used by readers during evaluation of statistical inference.

---

## Author Comment (AC2)

**Replies to Referee 2:**

Our replies are embedded in the referee's comments.

We wish to thank the referee for the useful insights offered.

**Comments by Referee 2:**

The paper A climate driven, altitudinal transition in rock glaciers dynamics detected through integration of geomorphological mapping and InSAR-based kinematics (authors: Bertone et al.) presents a rock glacier inventory in the western South Tirol established through a combined geomorphic approach and InSAR analysis. The research underscores the significance of InSAR in inventorying rock glaciers and assessing their surface kinematics. The paper is clearly structured and well-illustrated, containing sufficient critical reflections and arriving at appropriate conclusions. The methodology is rigorous and consistent, integrating several complementing techniques, whit data supporting the interpretations. The results align with the findings and there are no factual errors. Therefore, I recommend accepting the manuscript with minor revisions. These are outlined below.

It is not clear if for this study area previous rock glacier inventories were compiled. Please, clearly refer to this issue in the Introduction and refer to the nearby inventories.

We agree. The geomorphologic inventory for south-western South Tyrol presented here relies on a broader regional effort compiled between 2018 and 2020. When we submitted the original manuscript, the publication of the geomorphologic inventory of South Tyrol was under preparation. Now it is in press as a full paper in the context of ICOP 2024 (Scotti et al., in press).

Scotti R, Mair V, Costantini D, Brardinoni F. In press. A high-resolution rock glacier inventory of South Tyrol. Evaluating lithologic, topographic, and climatic effects. Full paper, 8 pp. ICOP 2024, 16-20 June 2024, Whitehorse, Yukon, Canada.

Now that the inventory is in press, we have added reference to the geomorphologic inventory in line 124-125:

"The geomorphologic rock glacier inventory in south-western South Tyrol is part of a broader mapping effort conducted across the entire Autonomous Province of Bozen/Bolzano (Scotti et al., in press)."

Line 126: what type of optical imagery you used?

We agree. Web links to arial photo-imagery and LiDAR-derived hillshade raster were provided in the captions of Figures 3, 5, 12 and 13. To address the reviewer's concern we have added such information in section 3.1. Accordingly, the revised sentence in lines 125-128 now reads as follows: "The identification, mapping and dynamic classification of each rock glacier relies primarily on the visual interpretation of 0.2-m gridded, optical imagery (i.e., orthophoto mosaics flown in 2014, 2017, and 2020) and a 2.5-m LiDAR-derived hillshade raster (i.e., 2006) -- all available as WMS resources at the Geological Web Portal of the Autonomous Province of Bolzano (https://geoportale.retecivica.bz.it/geodati.asp) -- and where available, on information drawn from local reports on road closure or damage to infrastructure associated with the advance of rock glacier fronts."

4b: the bars should correspond to altitude intervals not to altitude values. Please correct the elevation values below the graph either by shifting all the values to the left, or replace it with elevation intervals.

Thank you for this comment. We had not realized this problem with Figure 4b. We have modified the Y axis accordingly. Please see figure below.

[Figure]

Line 298: Can you shortly present here the elevation range of all the moving areas?

We agree. The range and median elevations are presented in the following new sentence, added after line 298: "They range in elevation from 1935 m to 3177 m a.s.l., and display a median of 2680 m a.s.l."

Based on the InSAR analysis 14 new rock glaciers were detected (fig. 6). I think it would be useful to shortly presents the elevation range of these rock glaciers.

We agree. The newly detected intact rock glaciers do not appear to be located at particularly low elevations: between 2440 and 3120 m. We have added the following sentence in line 300: "The latter landforms range in elevation between 2440 and 3120 m a.s.l.".

Fig 6: 144 rock glaciers could not be classified, which means around 18% of the total inventory. For these rock glaciers (RG) you`ve used the geomorphologic criteria, which seems logical. However, it might be necessary to address this issue in the Discussion section, particularly when discussing the limitations. Perhaps, the notion to convey could be that the geomorphic interpretation should not be entirely supplanted by InSAR; instead a combination of both approaches is deemed desirable.

We agree with the referee that geomorphic interpretation should not be entirely replaced by InSAR characterization. This notion is incorporated in the title (i.e., integration of geomorphological mapping and INSAR-based kinematics), in the methodology (i.e., we build an "integrated inventory", as well as in the opening paragraph of section 5.1 (Lines 421-429):

"In this contribution, we illustrate the importance of **complementing** a geomorphologic rock glacier inventory with InSAR-derived kinematic information. **Integration of the two inventorying approaches**, each of which characterized by specific strengths and weaknesses, allows reducing the overall uncertainty associated with the procedures of rock glacier detection, mapping and dynamic classification. On one hand, visual interpretation of multi-temporal optical imagery is crucial for outlining the morphological footprint of a rock glacier, and serves as a benchmark for developing automated mapping routines (e.g., Robson et al., 2020; Reinosch et al., 2021); on the other hand, visual interpretation of multi-temporal wrapped interferograms – except where decorrelation occurs – ultimately determines whether a rock glacier exhibits surface deformation, at what mean annual rate, and in which portion of its morphological footprint, thus confirming or rectifying prior evaluation solely based on the interpretation of morphological attributes.".

To address the referee's comment and reinforce this notion, we have incorporated the referee's sentence at the end of section 5.1 (lines 482-483) as follows: "Based on the foregoing quantitative evaluations, we suggest that geomorphic interpretation should not be entirely supplanted by InSAR-based kinematic characterization; instead, a combination of both approaches is deemed desirable".

Lines 348-349: You mention that the upper altitudinal limit of slow-moving areas (1-3 cm) are 300 m below the other faster classes. However, in Fig. 7a, this distinction is not very clear (please correct me if I`m mistaken). It appears that the upper green dots fall between 3000 and 3200 m, similar to the other graphs.

In 7b, c and d the upper dots occur a bit higher, but certainly not with a difference of 300 m. Please double-check!

We agree with the referee. Our description was not appropriate, since the upper altitudinal limit of 1-3 cm yr-1 moving areas drops by about 300 m across all but the southerly facing aspects (i.e., the cluster of 7 moving areas comprised between SE and SW aspects) that essentially share the same upper limit with the faster ones. Following this appraisal, we have modified lines 348-349 as follows: "Overall, the upper altitudinal limit of areas moving at 1-3 cm yr-1 plots 200-to-300 m below that of the faster classes (i.e., except for a cluster of 7 moving areas on southerly aspect; Figure 7a), which, by contrast, do not show altitudinal segregation from each other".

Additionally, high-velocity MA`s (> 1m) do not seem to occur at very high elevations. There are several RG below 2600 m which move very rapidly (which might be interesting to investigate further in the future), even if they fall below the elevation band 2600-2800 m, where the majority of the intact rock glaciers are found. However, it seems that very fast rock glaciers are not strongly constrained by elevation. An analysis over a more extended period of time might help to understand their recent behavior.

Thank you for the insight on the > 1m MAs. We think it is important to incorporate a comment on this in the manuscript. We do this in lines 348-349: "This is particularly the case of areas moving > 1 m yr-1, which cluster on easterly and westerly aspects, and appear to be not strongly constrained by elevation."

As for "very fast" rock glaciers, none of these have been detected in the study area within the 2018-19 period – the highest category represented being rock glacier moving annually from dm to m – and therefore it would seem difficult to draw conclusions on them. We agree that further analysis on extended baselines might be an interesting future task to pursue.

For completeness, below we report a table showing the descriptive statistics of rock glaciers stratified by kinematic classes, in which although not strongly constrained by elevation, we observe consistent increase of median and minimum elevation across progressively faster classes. We think that this progression agrees with the representation and the main message conveyed in Figure 15.

| RG vel class | n. obs. | Elevation (m asl) | | | |
| --- | --- | --- | --- | --- | --- |
| | | average | median | min | max |
| cm yr-1 | 145 | 2460 | **2475** | 1883 | 3102 |
| cm to dm yr-1 | 63 | 2630 | **2625** | 2209 | 3043 |
| dm yr-1 | 68 | 2660 | **2720** | 2104 | 3101 |
| dm to m yr-1 | 64 | 2660 | **2770** | 2334 | 3030 |

Line 365: I think it might be useful to create a graph similar with 4b for intact/relict RG after considering the velocity because otherwise, it might be confusing. Ultimately, the rock glacier inventory based on InSAR data represents the final version of this work. Upon examining these figures, it becomes apparent that no intact RG occurs below 2200 m (fig. 4b), whereas in 8b there are MAs below 2000 m. These low-altitude permafrost sites might also be an interesting finding of this work, so I recommend briefly referring to this in the Discussion (similar sites are found in various locations below 2000 m across mid-latitude mountains). Earlier, I asked you to present the elevation range of the 14 new rock glaciers detected because I am curious to know if these occur particularly at lower elevations.

Thank you for the useful comment. On the premise that Figure 4b shows "intact" rock glaciers solely classified through visual interpretation of morphological expression, InSAR-based dynamic reclassification of rock glaciers shows that these rock glaciers exist below 2000 m asl elevation band (i.e., a handful of RGs

that "become intact" in Figure 11b, all facing dominantly north). To highlight the presence of these low-lying landforms we have added the following text (lines 478-479): "It is noteworthy to highlight that dynamic reclassification allowed uncovering a cluster of low-lying rock glaciers (i.e., below 2000 m a.s.l.), dominantly facing north. This finding agrees with previous reports on the occurrence of intact rock glaciers around (or slightly above) 2000 m a.s.l. in other mid-latitude mountain settings, such as the Southern Carpathians (Vespremeanu-Stroe et al., 2012; Necsoiu et al., 2016)."

Following the referee's suggestion, we have also added the InSAR-based version of Figure 4b, here labelled Figure 11c.

[Figure]

Figure 11 …. (c) Total rock glacier area across altitudinal bands, stratified into moving and not moving categories after InSAR analysis. This representation excludes undefined landforms. In panel c, note generalized altitudinal overlap between moving and not moving rock glaciers (cf., Figure 4b).

As a consequence of this new figure, existing sentences in lines 474-478 have been modified as follows: "… thus indirectly weakening the altitudinal separation previously observed between morphologically intact and relict landforms both in terms of median front elevation across aspects (Fig. 4a) and total rock glacier area distribution (Fig. 4b). In particular, we observe that the median front elevation of rock glaciers that "become intact" (2375 m a.s.l.) approaches that of counterparts that "remain relict" (2270 m) (Table 4); vice versa, the median elevation front of those that "become relict" (2725 m) closely matches that of counterparts that "remain intact" (2675 m) (Table 4). With respect to total rock glacier area, the generalized altitudinal overlap between moving (i.e., intact) and not moving (i.e., relict) is striking, when compared to the corresponding output of the geomorphologic inventory (cf., Fig. 4b and Fig. 11c)."

As for the newly detected rock glaciers, they do not sit at particularly low elevations: between 2440-3120 m asl.

Vespremeanu-Stroe, A., Urdea, P., Popescu, R., Vasile, M. 2012. Rock Glacier Activity in the Retezat Mountains, Southern Carpathians, Romania. Permafrost and Periglacial Processes, 23, 127-137. https://doi.org/10.1002/ppp.1736.

Lines 379-380: you are correct, but the number of RG above 2900 m is also significantly low and this should be taken into consideration as well.

True, but still, out of 30 rock glaciers located > 2900 m asl (and that could be kinematically classified via InSAR), 22 have total moving area > 50% (they plot between the 1:2 and the 1:1 line). In this case, we think that our statement remains appropriate and not misleading.

To acknowledge the limited number of rock glaciers involved, we have rewritten the sentence as follows: "At elevations above 2900 m a.s.l., where clustering is highest (despite the limited number of observations involved) and data trend parallel to isometry (1:1 line), total moving area increases at about the same rate of rock glacier size."

Lines 393-394: This is an interesting finding, but keep in mind that the most extended RG in generally are not necessary moving the fastest presently. Additionally, in this study, I don`t believe the fastest rock glaciers are necessarily the most extended (please correct me if I`m mistaken). As you are aware, there are other factors influencing their extent, such as the duration of activity, contributing area, lithology and structural conditions etc.

We totally agree with the referee. Our representation and accompanying text refer to the percent surface of a rock glacier that is actively moving, regardless of rock glacier size. Indeed, the clustering trend being parallel to 1:1-1:2 lines confirms that rock glacier size does not matter. To make this point more explicit and avoid possible misunderstanding, we have added the following short bit to the sentence in line 393: "… irrespective of rock glacier size".

Figure 15: This is a very useful graph, but what is somehow surprising is the big number of intact RG occurring at MAAT above -1o Maybe would be interesting to add other few isotherm with a different color and to discuss on the reliability of this theoretical threshold (-1o C or -2o C) for discontinuous permafrost, respectively to refer to a threshold for sporadic permafrost in Tirol because it seems that there are enough moving areas/intact rock glaciers in areas with positive MAAT.

Although the figure now looks a little busier (i.e., the new isotherms overlap with the slope whiskers). We have modified Figure 15 adding isotherms 0, +1, +2 and +3 °C. We agree with the referee. To recap, we believe that: (i) the inferred altitudinal threshold for discontinuous permafrost is depicted by the inflection in MA percent cover and RG annual velocity functions at 2600-2800 m asl; (ii) an altitudinal threshold for sporadic permafrost may be inferred at 1880-2000 m asl; and (iii) a number of rock glaciers hold moving areas at positive MAAT values.

The following text was added in lines 578-580: "Interestingly, this representation aids elucidating that a number of moving areas, and relevant hosting intact rock glaciers, are located at positive MAAT values, and that locally a lower limit for sporadic permafrost may be set around the 1880-2000 m elevation band (i.e., + 3°C MAAT)."

Similarly, the sentence in lines 603-604 (section 6) was integrated as follows:

"We find that InSAR-based dynamic reclassification of rock glaciers: (i) aid detecting a cluster of low-lying intact rock glaciers (i.e., < 2000 m a.s.l.) associated with positive MAAT values; and (ii) induces substantial increase in the altitudinal overlap between relict and intact rock glaciers across slope aspects."

[Figure]

Lines 587-588: it`s hard to determine the importance of erosion rates for the acceleration of RGs in South Tirol. The reality is that using only a 1-year interval of velocity measurements makes it challenging to draw definitive conclusions. This could be related to various factors such as: snow cover regime, characteristics of the zero curtain, the freezing period in the active layer, intense summer rainfalls among others.

In our view, the main point is not the 13-month time window, as we believe that the trend shown in Figure 15 (based on hundreds of rock glaciers) is going to be relatively stable/resilient, even when using a 4-to-5 yr interval of InSAR data (as opposed to the temporal variability in the kinematics of a single, or a handful of rock glaciers); rather, as pointed out by the reviewer, the number of other factors potentially involved that might contribute to the altitudinal kinematic transition observed.

In this context, our statement wishes to make no inference on long term erosion rates associated to rock glaciers in SW South Tyrol, it plainly aims at highlighting similar altitudinal increase in RG kinematics (Fig 15) and rock wall erosion rates as constrained via cosmogenic nuclides by Draebig et al (2022). However, we also state that the extent to which the RG kinematic altitudinal increase may be due to frost cracking/rockfall rates feeding the rock glaciers -- besides permafrost occurrence alone -- is clearly beyond our reach with available data (lines 586-589).